# Insights into the Synergistic Antibacterial Activity of Silver Nitrate with Potassium Tellurite against *Pseudomonas aeruginosa*

Ali Pormohammad,[a,b] Andrea Firrincieli,[c] Daniel A. Salazar-Alemán,[a] Mehdi Mohammadi,[a] Dave Hansen,[a] Martina Cappelletti,[d] Davide Zannoni,[d] Mohammad Zarei,[e,f] Raymond J. Turner[a]

[a]Department of Biological Sciences, Faculty of Science, University of Calgary, Calgary, Alberta, Canada
[b]CCrest Laboratories, Inc., Montreal, Quebec, Canada
[c]Department for Innovation in Biological, Agro-Food and Forest systems, University of Tuscia, Viterbo, Italy
[d]Department of Pharmacy and Biotechnology, University of Bologna, Bologna, Italy
[e]Renal Division, Brigham & Women's Hospital, Harvard Medical School, Boston, Massachusetts, USA
[f]John B. Little Center for Radiation Sciences, Harvard T. H. Chan School of Public Health, Boston, Massachusetts, USA

Ali Pormohammad and Andrea Firrincieli are co-first authors. Order of authors chosen based on who initiated the project.

**ABSTRACT** The constant, ever-increasing antibiotic resistance crisis leads to the announcement of "urgent, novel antibiotics needed" by the World Health Organization. Our previous works showed a promising synergistic antibacterial activity of silver nitrate with potassium tellurite out of thousands of other metal/metalloid-based antibacterial combinations. The silver-tellurite combined treatment not only is more effective than common antibiotics but also prevents bacterial recovery, decreases the risk of future resistance chance, and decreases the effective concentrations. We demonstrate that the silver-tellurite combination is effective against clinical isolates. Further, this study was conducted to address knowledge gaps in the available data on the antibacterial mechanism of both silver and tellurite, as well as to give insight into how the mixture provides synergism as a combination. Here, we defined the differentially expressed gene profile of *Pseudomonas aeruginosa* under silver, tellurite, and silver-tellurite combination stress using an RNA sequencing approach to examine the global transcriptional changes in the challenged cultures grown in simulated wound fluid. The study was complemented with metabolomics and biochemistry assays. Both metal ions mainly affected four cellular processes, including sulfur homeostasis, reactive oxygen species response, energy pathways, and the bacterial cell membrane (for silver). Using a *Caenorhabditis elegans* animal model we showed silver-tellurite has reduced toxicity over individual metal/metalloid salts and provides increased antioxidant properties to the host. This work demonstrates that the addition of tellurite would improve the efficacy of silver in biomedical applications.

**IMPORTANCE** Metals and/or metalloids could represent antimicrobial alternatives for industrial and clinical applications (e.g., surface coatings, livestock, and topical infection control) because of their great properties, such as good stability and long half-life. Silver is the most common antimicrobial metal, but resistance prevalence is high, and it can be toxic to the host above a certain concentration. We found that a silver-tellurite composition has antibacterial synergistic effect and that the combination is beneficial to the host. So, the efficacy and application of silver could increase by adding tellurite in the recommended concentration(s). We used different methods to evaluate the mechanism for how this combination can be so incredibly synergistic, leading to efficacy against antibiotic- and silver-resistant isolates. Our two main findings are that (i) both silver and tellurite mostly target the same pathways and (ii) the coapplication of silver with tellurite tends not to target new pathways but targets the same pathways with an amplified change.

Address correspondence to Raymond J. Turner, turnerr@ucalgary.ca, or Andrea Firrincieli, andrea.firrincieli@unitus.it.

The authors declare no conflict of interest. Pormohammad A, Turner RJ. February 23, 2022. US Patent No. 63/312,861 filed with WIPO international. PCT/IB2023/051621, Metal(loid)-based compositions and uses thereof against bacterial biofilms.

**KEYWORDS** *Pseudomonas aeruginosa*, mechanism, metabolomics, silver, synergy, tellurite, transcriptomics, RNA sequencing, *Caenorhabditis elegans*

Antimicrobial resistance (AMR) to conventional antimicrobial agents has become one of the most concerning health crises (reviewed in references 1 and 2). According to a recent publication by *The Lancet*, 4.95 million deaths worldwide were associated with AMR bacteria, and 1.27 million were attributed directly to AMR in 2019 (3). The decreasing effectiveness of existing antibiotics has made it crucial to develop novel antibacterial agents (4). Various novel approaches are being pursued to tackle the AMR problem from antimicrobial peptides to phages (5, 6). The application of potent antibacterial agents synergistically to formulate a novel effective combination(s) is promising and has interested us for several reasons. First, it can lead to the complete eradication of cell viability and the prevention of bacterium recovery (7). Second, mixed antimicrobials with different cell targets decrease future resistance development, since it is astronomically rare to evolve resistance simultaneously toward two different antimicrobials in a single cell at the same time (7, 8). Third, the use of mixed antimicrobials permits the reduction of the effective concentration of the antibacterial agents, which leads to reduced toxicity and side effects for the host (7, 9–11).

We see promise in the use of metal/metalloid-based antimicrobials (MBAs) (8, 12, 13). We have shown that metal-metalloid(s) can be synergistic with antiseptics (14), as well as with each other (7, 8), and can support convention therapy. Our recent studies showed the strong synergistic antibacterial and antibiofilm activity of silver nitrate ($AgNO_3$) with potassium tellurite ($K_2TeO_3$) against both Gram-positive and Gram-negative bacteria (7, 8). The metal-metalloid [metal(loid)] silver-tellurite combination had the most effective bacteriostatic and bactericidal synergism among 5,760 different MBA combinations tested against planktonic cells (7) and showed biofilm prevention and eradication synergistic activity among the 1,920 combinatorial MBAs tested against *Pseudomonas aeruginosa* (8).

Tellurium is a metalloid and is found to be antimicrobial in the IV oxidation state ($TeO_3^{2-}$), and it tends to be the most bacteriostatic and bactericidal of all the MBAs, with low effective concentrations against a large number of bacterial species, particularly Gram-negative bacteria (7). Our study validated this, since it shows that tellurite has the most effective antibiofilm activity, among nine MBAs, against clinical isolates of *P. aeruginosa* (8). Moreover, Te(IV) has synergistic antibacterial activity with other antibiotics such as tetracycline, ampicillin, cefotaxime, and chloramphenicol against *P. aeruginosa* and *Escherichia coli* (10). To date, although considerable effort has been put toward the molecular mechanism of tellurite antibacterial activity (15–24), there are still knowledge gaps and inconsistencies (21, 23). A cell system view of the molecular mechanisms and specific targets could help to provide information on how to use tellurite effectively and extend its industrial and clinical applications (25, 26). Oxidative stress is the primary reported activity for the toxicity of tellurite (15–20, 27), and yet there is still suspicion that other mechanisms underlie this oxidative stress observations (21, 23).

The coinage metal silver has been traditionally used for the treatment of different infections (28). The literature suggests that a wide variety of molecular mechanisms of toxicity has been suggested for silver exposure to bacteria in the literature (11, 12, 29–36). These include [Fe-S] center disruption, reduced thiol oxidation, binding to and inactivation of specific enzymes, reactive oxygen species (ROS) production, and membrane damage. Silver has also been reported to have synergistic antibacterial activity with different organic antimicrobials (30, 34, 37), and yet the explanation of the synergies has not been explored in depth.

In this study, we aimed to define the molecular mechanisms of the effect of combined exposure to silver with tellurite and obtain an explanation for this combination's superior synergism as antimicrobial MBAs. After 2 h of exposure of a *P. aeruginosa* planktonic culture at the MIC(s) of each agent used individually and/or in combination, we adopted transcriptomic and metabolomic approaches to obtain a system biology view of the global changes of a *P. aeruginosa* culture grown in simulated wound fluid to silver and tellurite either alone or added together in combination. The data were complemented by direct biochemical

assays in which ciprofloxacin and gentamicin were used as antimicrobial comparators. We also evaluated the stress of these MBAs on the animal model system *Caenorhabditis elegans* toward an initial evaluation of any adverse effects to a eukaryote system.

We present here novel insights into the mechanisms of these two metal-based antimicrobials and show that they work synergistically through magnifying each other's antibacterial effects, and we also provide evidence for their safe use.

## RESULTS

**Bacteriostatic and bactericidal activity of a silver-tellurite combination against clinical antibiotic resistance isolates.** The bacteriostatic (MIC) and bactericidal (minimal bactericidal concentrations [MBC]) activities of silver, tellurite, and silver-tellurite combinations were explored against clinical isolates to identify the efficacy profiles in comparison with the commonly used antibiotics gentamicin and ciprofloxacin against *P. aeruginosa*. These experiments were performed in simulated wound fluid (SWF) under aerobic conditions to verify the silver-tellurite antibacterial activity in a clinical setting's more relevant physiological state.

Of the 39 clinical isolates of *P. aeruginosa*, 17 (43%) displayed resistance to ciprofloxacin or gentamicin. Of these 17 resistance isolates, 16 (94%) isolates were resistant to gentamicin, while 1 (5.8%) isolate was resistant to ciprofloxacin; 9 (53%) isolates were isolated from burn wounds, and 8 (47%) were from cystic fibrosis (CF) patients. No isolates were resistant to both antibiotics (Fig. 1).

Resistance levels against silver and antibiotics (especially gentamicin) in clinical isolates were higher than for the reference strains PAO1 and ATCC 27853. For instance, the silver MIC for the reference strains was 0.125 mM, while the MIC was a full log higher when tested against clinical isolates with an average of 1.25 mM (range, 0.62 to 2.5 mM) when cultured in SWF. Common usage of silver for various consumer goods and clinical approaches might be the most probable explanation for this observation. The same result was found in our previous study for *P. aeruginosa* grown as a biofilm (8).

Therefore, clinical isolates acquire a level of tolerance not only to antibiotics but also now to silver, which is commonly used in wound care. Conversely, clinical isolates were comparably more susceptible than reference strains when exposed to tellurite, which might be due to the limited application of tellurite in health care. All clinical isolates were highly susceptible to the silver-tellurite combination, which suggested that they act synergistically (Fig. 1). Detailed synergism antibacterial data of the silver-tellurite combination is given in Table S1 and Fig. S1 and S2 in the supplemental material for both bacteriostatic and bactericidal efficacies. The silver-tellurite combination showed remarkable synergistic antibacterial activity against all clinical isolates. FIC (fractional inhibitory concentration) scores ranged from 0.51 to 0.16 and FIB (fractional eradication concentration) from 0.62 to 0.157, defining synergy for the silver-tellurite combination against all clinical isolates (synergy calculations and data are given in Table S1 and other supplemental material). Remarkably low concentrations of silver (0.039 mM) and tellurite (0.016 mM) applied in combination were required for a bacteriostatic effect against 12 of 18 clinical isolates compared to the MIC of each of the two metal(loids) when supplied alone (silver at 1.25 mM [isolate range from 0.62 to 2.5 mM] and tellurite at 0.125 mM [0.062 to 0.25 mM]) in simulated wound media (Fig. 1).

**Differential gene expression profile in *P. aeruginosa* exposed to silver, tellurite, and silver-tellurite.** Evaluation of the global transcription from exposure to silver (0.125 mM AgNO$_3$), tellurite (0.25 mM K$_2$TeO$_3$), and silver-tellurite (0.125 mM AgNO$_3$ and 0.25 mM K$_2$TeO$_3$ added together) (see Table S2) of a culture of *P. aeruginosa*, strain ATCC 27853, in mid-log phase for 2 h led to 124, 109, and 436 differentially expressed genes (DEGs), respectively (Fig. 2A; see also Table S3). Principal-component analysis on r-log-transformed expression data showed that samples cluster very well by challenge condition, with a clear separation along the first component of those representing silver-tellurite-treated cells (see Fig. S3). Despite the large differences observed along the first component, genes involved in sulfur metabolism (*ssu*) and uptake (*sbp*) were identified among the top 10 upregulated genes upon exposure to silver, tellurite, and silver-tellurite (see Fig. S4).

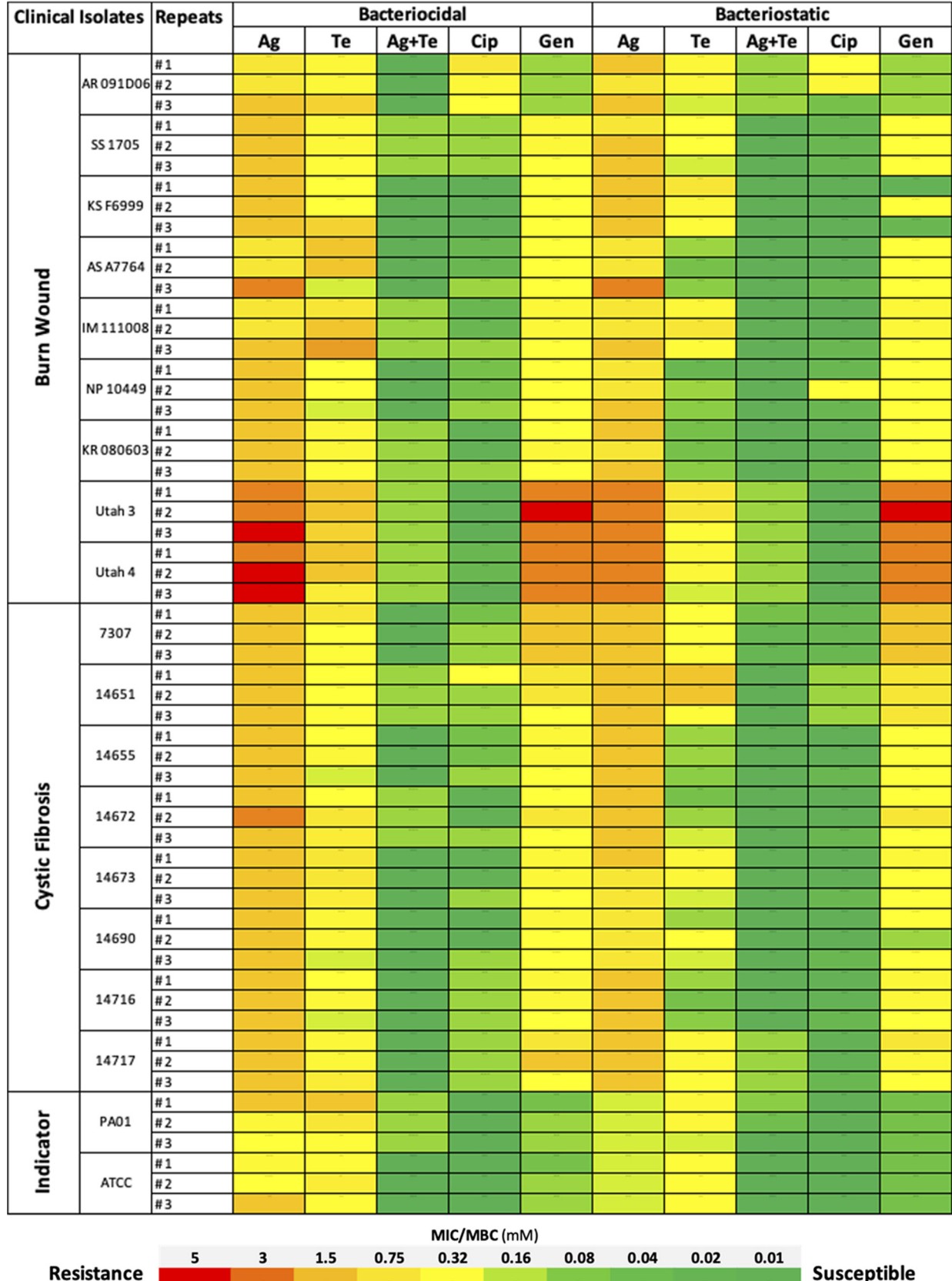

**FIG 1** Antibacterial effectiveness of Metal(loids) combination in comparison to antibiotics against clinical isolates of *P. aeruginosa*. MICs and MBCs of silver nitrate (Ag), potassium tellurite (Te), and Ag-Te combinations against clinical isolates in comparison to gentamicin and ciprofloxacin in the SWF. Detailed synergism information is shown in Fig. S1 and S2 and in Table S1 in the supplemental material. Cip, ciprofloxacin; Gen, gentamicin; Ag, silver nitrate; Te, potassium tellurite.

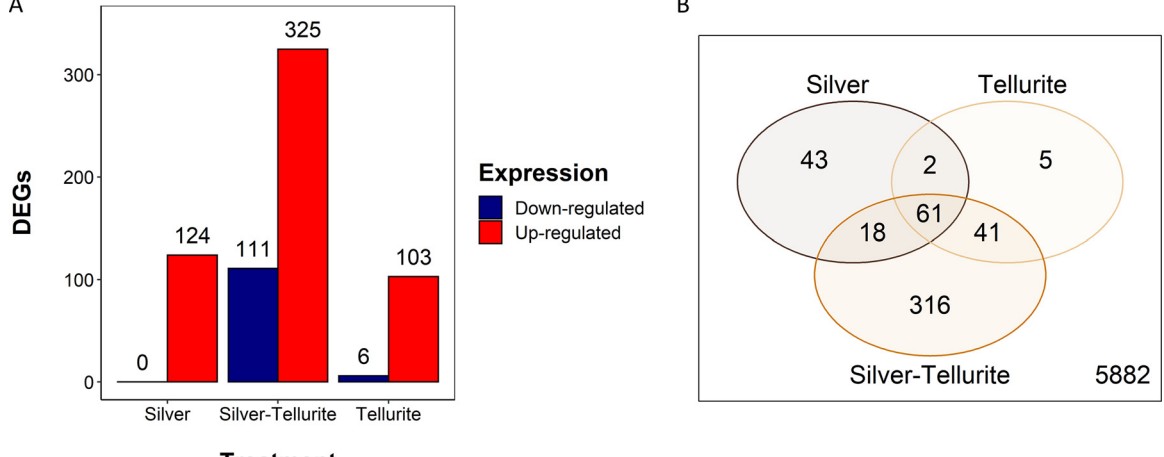

**FIG 2** DEGs in *P. aeruginosa* ATCC 27853 exposed to silver, tellurite, and silver-tellurite. (A) Bar plots of up- and downregulated genes in silver-, tellurite-, and silver/tellurite-exposed cells compared to the control condition (i.e., the absence of metal). Only genes with a false discovery rate of <0.05 and a |log$_2$-fold change| of <1.0 are plotted. (B) Venn diagram of DEGs identified under all three conditions.

In the silver- or tellurite-exposed cultures almost all the DEGs were upregulated, while for exposure to silver with tellurite, 325 and 111 genes were significantly up- and downregulated, respectively (Fig. 2A). Overall, a total of 61 DEGs were common to all the tested conditions, while 43, 5, and 316 DEGs were uniquely induced by silver, tellurite, or silver-tellurite, as illustrated in a Venn diagram (Fig. 2B).

Functional overrepresentation analysis was performed on these unique DEGs to define the metal-specific global transcriptional changes in the *P. aeruginosa* culture. Given the reduced number of these unique genes, no specific functional categories (GO terms) were overrepresented in the pool of DEGs specifically induced by each of the two metal (loids). The genes upregulated by tellurite were *ppk2* (polyphosphate kinase 2), *yggL* (50S ribosome-binding protein), and *metH* (methionine synthase), while the two downregulated genes were *aldB* (aldehyde dehydrogenase) and *pedF* (cytochrome *c*-550) (Fig. 3A). Similarly, although no specific functional categories were significantly over-represented within the pool of DEGs uniquely induced by silver, several of these DEGs were organized in gene clusters coding for different classes of ABC transporters and multidrug Resistance-Nodulation-Division (RND)-export systems, including gene clusters for the components of the spermidine/putrescine ABC transport system, the hydroxyproline ABC transport system, the gold/copper resistance efflux system, and copper/silver efflux (Fig. 3B). With a total of 316 condition-specific genes, the silver-tellurite combination was the condition with the largest number of DEGs. Enrichment analysis of the silver-tellurite DEGs identified a total of six enriched functional categories (GO terms) all related to the uptake of branched-chain amino acids and the response to various cellular stimuli (Fig. 3C; see also Table S4), including genes coding for molecular chaperones, protein folding, and heat shock proteins.

Finally, the 61 DEGs common to the silver, tellurite, and silver-tellurite conditions were significantly enriched in genes involved in the metabolisms of sulfur-containing compounds, i.e., alkane sulfonate, sulfur amino acids, and sulfate uptake (see Table S5). Specifically, the gene clusters *tauABCD* and *ssuEADCB*, which are required for the uptake and utilization of taurine and alkene sulfonates as sulfur sources, were strongly upregulated in all three challenges, along with the *cys* gene clusters involved in the assimilatory sulfate reduction (Fig. 3D). The upregulation of *cys*, *ssu*, and *tau* gene clusters suggests the presence of a sulfate starvation response because of exposure to silver and tellurite (38). The central role of sulfur metabolism changed upon exposure to silver and tellurite is also supported by the upregulation of the genes coding for the *dsb/ccm* system which controls the redox state of disulfide bonds of periplasmic proteins (39) (Fig. 3D). Interestingly, *dsbA* (A4W92_RS16195; see Table S3) and *dsbB* (A4W92_RS19415; see Table S3), which are also parts of the *dsb/ccm* system (39), resulted in overexpression only for the silver-tellurite condition, indicating that

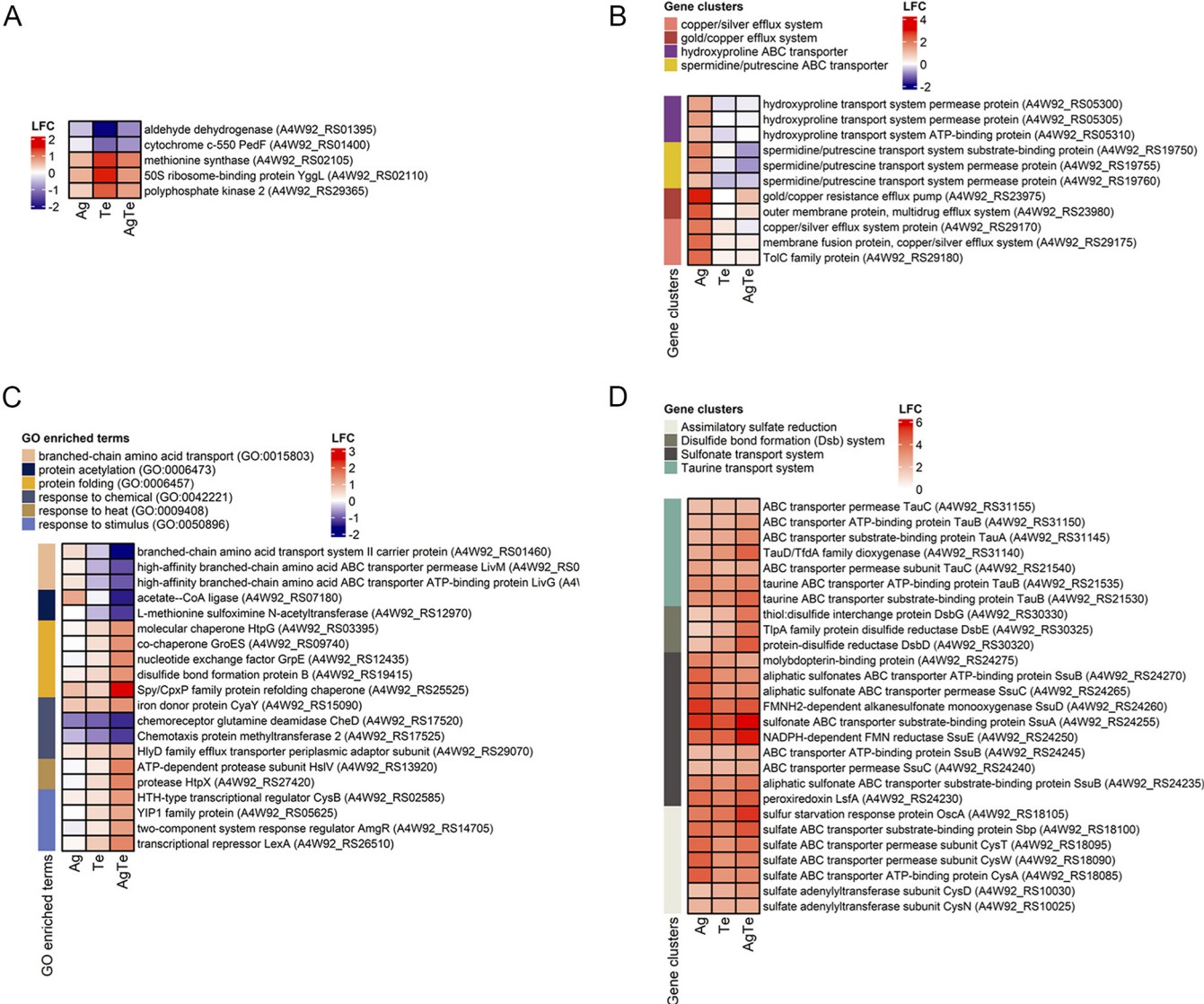

**FIG 3** Heatmaps of the log$_2$-fold change of differentially expressed stress response-related genes and gene clusters in *P. aeruginosa* ATCC 27853 exposed to silver, tellurite, and silver-tellurite. (A to C) DEGs unique to tellurite (A), silver (B), and silver-tellurite (C). (D) DEGs common to all three conditions metalloid exposures were at MICs of 0.125 mM silver nitrate (Ag), 0.25 mM potassium tellurite (Te), or 0.125 mM Ag plus 0.25 mM Te (for the Ag-Te combination).

members of the same system can be differentially regulated upon different degrees of metal (loid) stress.

**DEGs related to the synergistic effect induced by the copresence of tellurite with silver.** To investigate the additional effect of tellurite on the global transcriptional changes induced by silver, we analyzed the 163 genes that were detected in *P. aeruginosa* to be substantially differentially expressed only when treated with silver-tellurite but show only moderate to low expression changes (|log$_2$-fold change| < 0.5) in cells exposed to either silver or tellurite compared to the control (no metal).

Among these, genes showing either positive or negative changes were involved in multidrug resistance, the response to stressors [including toxic metal(loids) and aromatics], the response to cysteine oxidation, the response to unfolded proteins and protein degradation, and protein homeostasis (Fig. 4). Notably, among these stress-responsive genes, the superoxide oxidase (SOO) was significantly overexpressed in response to silver-tellurite, while it showed little or no changes in the presence of either the single tellurite or silver exposure. The SOO is a membrane-bound oxidase which contains a *b*-type diheme (40). SOO is expressed at a higher level during the exponential growth phase that can quench membranous generated superoxide (O$^-$) as a consequence of electron leakage from redox

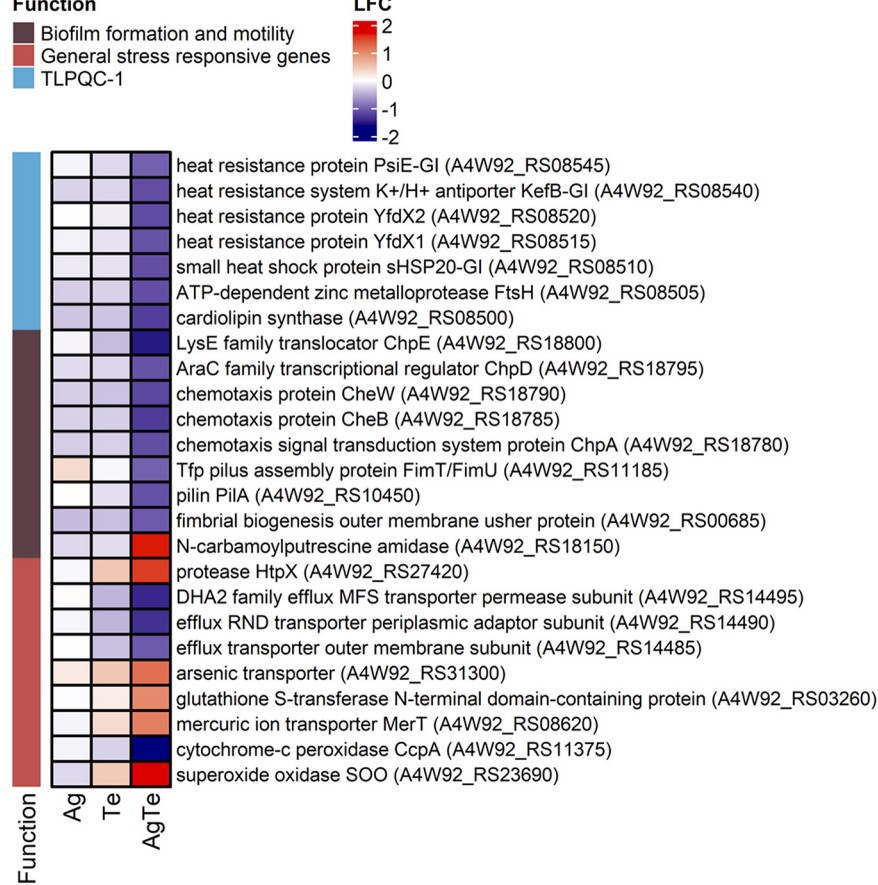

**FIG 4** Heatmaps illustrating log$_2$-fold change of stress response-related genes differentially expressed in *P. aeruginosa* ATCC 27853 exposed to silver, tellurite, and silver-tellurite. Only stress response-related genes showing moderate to low expression changes (|log$_2$-fold change| < 0.5) in silver- and tellurite-exposed cells are shown. Metalloid exposures were at MICs of 0.125 mM silver nitrate (Ag), 0.25 mM potassium tellurite (Te), or 0.125 mM Ag plus 0.25 mM Te (for the Ag-Te combination).

electron carriers of the respiratory chain. Compared to cytoplasmic superoxide dismutase (SOD), in SOO the superoxide is directly oxidized to molecular oxygen using ubiquinone as an electron acceptor (40). The strong upregulation of the SOO indicates that tellurite in combination with silver enhances the generation of superoxide at the level of the respiratory chain. Conversely, *P. aeruginosa* significantly downregulates the type 1 transmissible loci for protein quality control (TLPQC-1), whose gene products are involved in protein homeostasis and thermotolerance in response to antimicrobials (41, 42).

The addition of tellurite to silver also affects the expression of genes involved in biofilm formation and cell motility (Fig. 4). Regarding cell motility, the gene cluster coding for the chemosensory pili system proteins (Chp), and the genes coding for type IV fimbrial biogenesis (FimT/FimU) and pilin (PilA) proteins were significantly downregulated in silver-tellurite-treated cultures, while they were not differentially expressed in silver- and tellurite-exposed cells. Regarding biofilm formation, the *aguB* gene (*N*-carbamoyl putrescine amidase), which controls the intracellular levels of putrescine and c-di-GMP with a positive effect on biofilm formation, was significantly upregulated upon silver-tellurite exposure (43). The upregulation of genes promoting biofilm formation (*aguB*), in parallel with the downregulation of genes involved in type IV pilus-mediated twitching motility (*chp*), indicates that tellurite in combination with silver rearranges a transcriptional pattern associated with the transition from the planktonic to biofilm cell state/growth while limiting the bacterial twitching motility.

**Evaluation of extracellular metabolite levels in a culture of *P. aeruginosa* exposed to silver, tellurite, and silver-tellurite.** The metabolic boundary flux method was utilized to measure metabolites that are produced or consumed by bacterial cells exposed to a stress challenge within an *in vitro* cell culture, which can provide insight into the cell's physiological

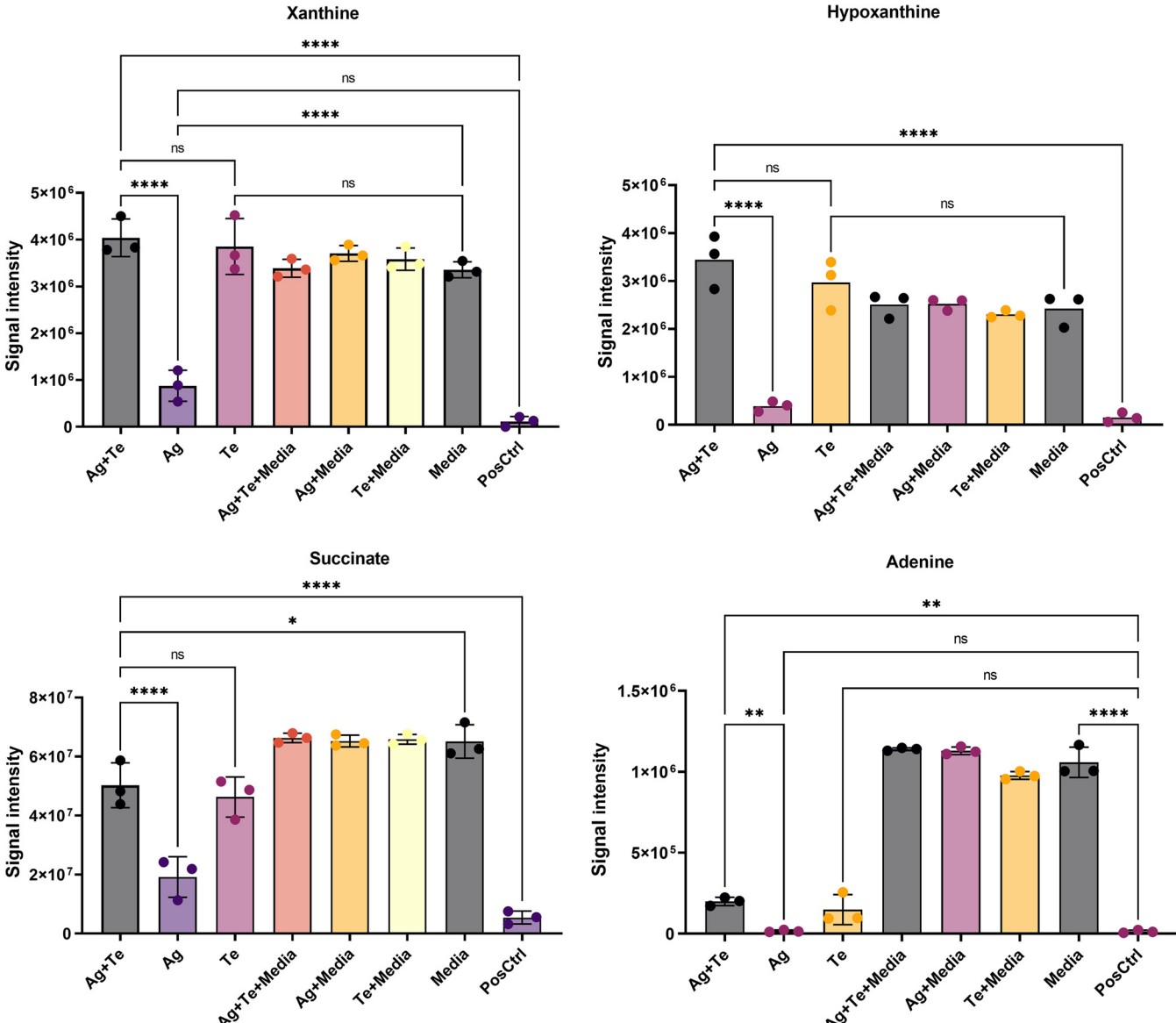

**FIG 5** Metabolomics investigation of *P. aeruginosa* extracellular metabolites after exposure to antimicrobials. The xanthine and hypoxanthine, succinate, and 10-hydroxydecanoic acid levels in *P. aeruginosa* ATCC 27853 treated with MICs of Ag, Te, and Ag-Te (0.125 mM silver nitrate [Ag], with 0.25 mM potassium tellurite [Te], or 0.125 mM Ag plus 0.25 mM Te [for the Ag-Te combination]) in comparison to the control groups (PosCtrl, treated with PBS. Controls of metal effects to medium metabolites are defined as Ag and/or Te + media). The statistical differences between treatments were calculated by one-way ANOVA with *post hoc* Tukey correction (*, $P < 0.05$; **, $P < 0.01$; ***, $P < 0.001$; ****, $P < 0.0001$).

fitness (44–46). This emerging method that focuses on the extracellular metabolites can capture microbe phenotypic information and provide information to understand the metabolism of cultured microbes. The method has provided insights into the metabolic flux profiles of antimicrobial susceptibility (44–46). Since the consumed or produced metabolites found in the extracellular environment are water soluble and less diverse compared to intracellular metabolites, this method offers an easy sample preparation procedure since it only requires the separation of cultured cells from the metabolites in the growth medium (46, 47).

Extracellular metabolomics identifies metabolites primarily related to energy homeostasis, amino acids metabolism, and nucleotide metabolism (see Fig. S5). Focusing on the metabolites showing the most significant differences between treatments (Fig. 5), we found that the energy-related metabolite succinate was consumed more in *P. aeruginosa* that was either not treated or silver treated, while less consumption was observed in cells exposed to silver-tellurite or tellurite alone. The latter observation is interesting and supports two further results under both tellurite challenges, namely, the downregulation of the E2 subunit

of pyruvate dehydrogenase (A4W92_RS13735; see also Table S3), which is likely to restrict glycolysis at the phosphoenolpyruvate (PEP)/pyruvate level, and the membrane uncoupling (see below) with consequent restriction in the activity of the dicarboxylic transporter (Dct) (48). On the other hand, the parallel upregulation of acyl coenzyme A (acyl-CoA) dehydrogenase (A4W92_RS03220; see Table S3) (which catalyzes the first step in the $\beta$-oxidation of fatty acids) might be an alternative pathway in the generation of acetyl-CoA to allow the operation of the tricarboxylic acid cycle since oxaloacetate would possibly be generated via carboxylation of PEP. These steps are examined in the Discussion in light of the upregulation of SOO (Fig. 4) and overreduction of the quinone pool (UQ pool) by SOO and ETF (electron transferring flavoprotein).

We focused our attention on succinate since it was reported as a preferred carbon source for *P. aeruginosa* before cells start using other carbon sources (49, 50), while glucose and amino acids are preferred carbon sources in other bacterial species (49, 50). Further, both carbon catabolite repression and high metabolic versatility of *P. aeruginosa* were proposed by Rojo (50) as the main determinants in the choice by the bacterium of new sources and in the expression of new virulence factors (50).

Purine metabolism was also identified here. Xanthine and hypoxanthine were significantly overproduced in the silver-tellurite- and tellurite-exposed groups, and consumption of these two metabolites is observed for untreated or silver-exposed groups. Xanthine was reported previously as a biomarker for the diagnosis of the presence of *P. aeruginosa* infections (51). Adenine was significantly consumed, particularly in the untreated and silver-treated groups and less so for the silver-tellurite- and tellurite-exposed groups. The breakdown of xanthine, hypoxanthine, and adenine in *P. aeruginosa* has been reported previously (51). Both *de novo* purine biosynthesis and salvage pathways have a critical role in biofilm formation (52) and cellular processes, including cell signaling, encoding the genetic constituents, and energy metabolism (53). In the *de novo* purine biosynthesis pathway, 5-phosphoribosyl-$\alpha$-diphosphate (PRPP) synthesizes IMP through 11 enzymatic steps and some of these reactions utilize ATP, followed by converting IMP to GMP and AMP and subsequently to signaling molecules of ppGpp(p) and c-di-GMP (54). In Fig. 4, we see a link of heightened DEGs related to the general stress response and biofilm formation/motility that rely on these signaling systems. Due to the high energy cost of the biosynthesis of purine nucleotides, the salvage pathway proceeds to obtain the purine bases from nucleic acid turnover, including hypoxanthine, adenine, and guanine, and convert them to IMP, AMP, and GMP, respectively (55).

**Assessments of the physiological response of *P. aeruginosa* exposed to silver, tellurite, and silver-tellurite.** Different studies have reported various mechanisms of antimicrobial action for silver or tellurite. Generally, the main reported mechanisms are thiol chemistry (16) and related oxidative stress (17) (through disrupting [Fe-S] clusters [18]), cell membrane dysfunction (at least for silver) (11), and changes to energy pathways (29, 34). Below, we look at our data more closely by adding additional assays to validate the metabolomics and transcriptomic findings.

**(i) Sulfur homeostasis.** For both the silver and the tellurite challenges, we see sulfur metabolism-related genes differentially expressed. These are genes that code for proteins involved in sulfur homeostasis such as the sulfur starvation response protein (*oscA*), the sulfate-binding protein (*sbp*), putative sulfonate monooxygenase (*ssuD* and *ssuE*), an FMN-dependent monooxygenase reductase for the utilization of aromatic sulfonates as sulfur sources, the sulfate/thiosulfate importer (*cysA*), and the *tau* gene involved in alkene sulfonate uptake (56). These genes are also involved in cysteine metabolism. A key cysteine containing peptide is glutathione, which is the major reduced thiol in cells and is key for maintaining cell redox poise (57). It is well accepted that tellurite leads to rapid glutathione oxidation and thus would deplete this key antioxidant (16, 17, 58). Silver also leads to reduced thiol oxidation through complexation (59). Our reduced thiol (RSH) assay showed that the loss of RSH content under the individually silver- or tellurite-exposed groups was surprisingly more than that of the silver-tellurite combination (see Fig. S6). This unexpected observation may be because of the upregulation of sulfur homeostasis genes helping to maintain a reducing environment in the cell. It is worth noting that a visible phenotype of tellurite exposure to cells is the culture becoming black due to the RSH-catalyzed Te(IV)

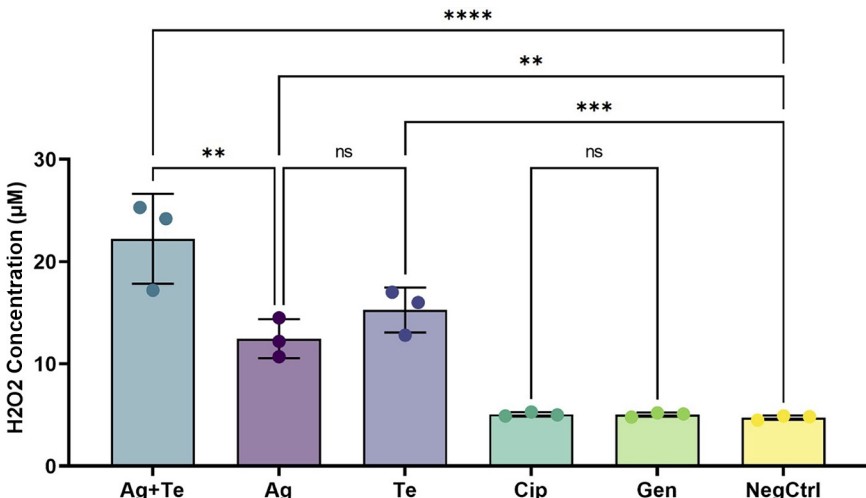

**FIG 6** Hydrogen peroxide ($H_2O_2$) levels in *P. aeruginosa* after exposure to antimicrobials. *P. aeruginosa* ATCC 27853 treated with MICs of Ag, Te, or Ag-Te (0.125 mM silver nitrate [Ag], with 0.25 mM potassium tellurite [Te], or 0.125 mM Ag plus 0.25 mM Te [for the Ag-Te combination]) antibiotic exposure at 1.25 $\mu$M gentamicin and ciprofloxacin in comparison to the control (NegCtrl; working reagent + untreated bacteria). The statistical differences between treatments were calculated by one-way ANOVA with *post hoc* Tukey correction, where $P < 0.05$ (for Ag+Te to Te) is annotated by asterisks (**, $P < 0.01$; ***, $P < 0.001$) ($n = 3$). Cip, ciprofloxacin; Gen, gentamicin.

reduction to Te(0), primarily from reduced glutathione (16, 17). The reduction leads to nano-precipitates in the cells (21), resulting in a black culture/colony phenotype. We observed this black phenotype with tellurite exposure; however, with the silver-tellurite combination, the culture was only light gray, reflecting that less RSH catalyzed reduction occurred, providing a visible phenotype agreement with the observations of Fig. S6.

**(ii) Peroxide production.** Oxidative stress or ROS is likely the most common mechanism(s) for most antibacterial agents (60), whether produced directly or indirectly. ROS can damage membrane lipids, DNA, and cellular proteins and thus directly leads to cell death (60, 61). Our transcriptome data show peroxiredoxin (A4W92_RS24230; see Table S3) expression is enhanced under all three MBA challenge conditions, and the gene expression level of superoxide oxidase is highly enhanced under silver-tellurite exposure. Peroxiredoxins are a ubiquitous family of cysteine-dependent peroxidase enzymes that play dominant roles in regulating peroxide ($H_2O_2$) levels in the cell as part of oxidative stress (62). Hydrogen peroxide is produced as a byproduct of oxidative respiration even under normal respiration conditions; thus, disconnection of the respiratory chain by silver and/or tellurite would lead to increased amounts of $H_2O_2$ in the cell. As such, peroxidase and superoxide oxidase would be key enzymes required to tolerate the metal ion challenge.

To complement our transcriptomics data, we measured the $H_2O_2$ concentrations via a colorimetric assay to detect $H_2O_2$ production after 2 h of exposure to the agents. It showed silver-tellurite-exposed bacteria producing higher $H_2O_2$ levels (24.5 $\pm$ 0.06 $\mu$M) compared to tellurite (16 $\pm$ 0.08 $\mu$M) ($P < 0.05$) or silver (12.2 $\pm$ 0.05 $\mu$M) ($P < 0.01$) alone compared to unchallenged levels (4.84 $\pm$ 0.05) (Fig. 6).

**(iii) Release of iron from proteins.** A challenge of silver, tellurium, zinc, mercury, cadmium, or copper has been reported to lead to breaking down iron-sulfur clusters to release $Fe^{+2}$ that then through Fenton reactions yield various ROS (18, 63, 64). Although no [Fe-S] chaperone or assembly genes were identified in our transcriptomic study, we decided to still explore this by performing a direct colorimetric assay using Ferene-S to measure free $Fe^{+2}$ after exposure of the *P. aeruginosa* culture with silver, tellurite, or their combination. The $Fe^{+2}$ concentration in 2-h-exposed bacteria was compared to a negative control (treated with PBS) and a positive control (10 min boiling to break the cluster). Ciprofloxacin and gentamicin were used as comparators, and their mechanism of antibacterial activity is not expected to affect [Fe-S] clusters in proteins. The comparative levels of free $Fe^{+2}$ in the metal-treated groups were higher than that of the unexposed control and ciprofloxacin/

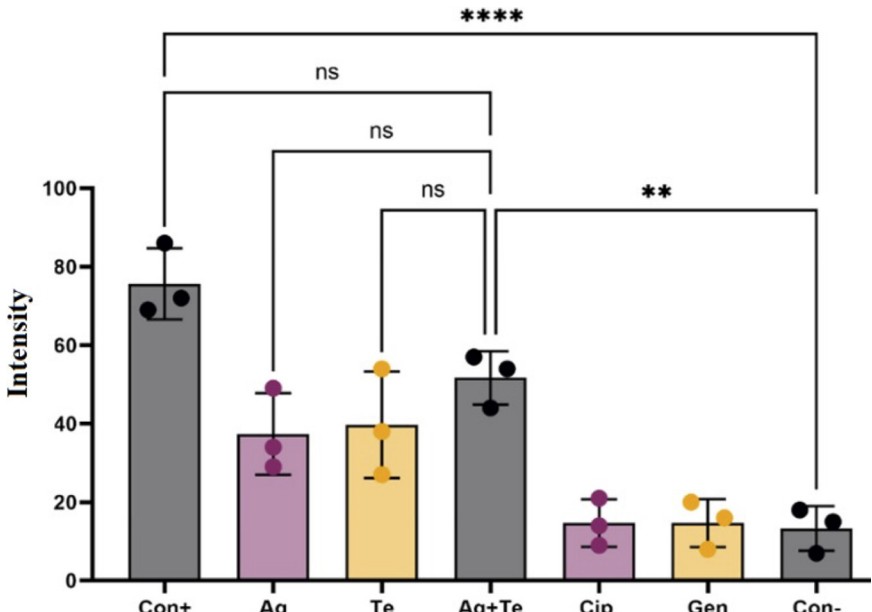

**FIG 7** Effect of antimicrobials on membrane integrity. The relative fluorescence intensity from the dye PI is shown ($n = 3$). Treatment with MICs of Ag, Te, or Ag-Te (0.125 mM silver nitrate [Ag], with 0.25 mM potassium tellurite [Te], or 0.125 mM Ag plus 0.25 mM Te [for the Ag-Te combination]) antibiotic exposure at 1.25 $\mu$M gentamicin and ciprofloxacin. The statistical differences between treatments were calculated by one-way ANOVA with *post hoc* Tukey correction, and asterisks indicate significance (*, $P < 0.05$; **, $P < 0.01$; ***, $P < 0.001$; ****, $P < 0.0001$). Cip, ciprofloxacin; Gen, gentamicin; Ag, silver nitrate; Te, potassium tellurite; Con+, control of bacteria boiled to disrupt the membrane; Con–, control of treatment with PBS.

gentamicin groups. We observed that the silver-tellurite combination-exposed group led to higher iron release levels than individual silver or tellurite treatments (see Fig. S7). This suggests that iron is potentially released from [Fe-S] and/or cytochrome centers by silver as suspected by others (18) and is likely responsible for the ROS produced that is leading to the need to overexpress oxidative stress response genes and is consistent with the observation of increased $H_2O_2$ levels. The results confirm that the silver-tellurite combination has a mechanism of antibacterial action similar to that of the individual metal(loids) but is much more enhanced.

**(iv) Membrane damage.** To survey bacterial membrane disruption after exposure to metal(loids), the probe propidium iodide (PI) was used. The polarity of PI allows it to penetrate only leaky cell membranes, which are characteristic of dead or damaged cells. Further, PI is a high-affinity nucleic acid dye that upon chelation leads to high fluorescence intensity. PI staining thus provides an indirect measure of its ability to enter cells due to a damaged membrane and/or devoid of an electrochemical potential (65).

After 2 h of exposure to metal(loids) (Fig. 7), the bacteria treated with either silver or tellurite, and especially the silver-tellurite combination, showed increased numbers of cells stained red, indicating damaged membranes in comparison to the untreated samples. Ciprofloxacin and gentamicin were used as controls that are not expected to target the cell membrane of bacteria. These data suggest that silver and tellurite affect the cell membrane as well, especially the silver-tellurite combination. Previous studies have shown that tellurite enters the cells via energetics provided by the $\Delta$pH component of the proton motive force (PMF) (66) but in a ratio of 1 nmol/5,000 cells, tellurite strongly damages the permeability of the plasma membrane (67). The damaged membrane would uncouple the PMF and thus lose the ability to efflux the PI by multidrug transporters (68, 69). Therefore, we can conclude that silver-tellurite are facilitating PMF uncoupling by either damaging the membrane and/or affecting the electron transfer chain, in addition to previous studies showing that tellurite (67, 70) and silver (11, 71, 72) strongly damage the cell membranes of bacteria.

**Evaluation of beneficial/harmful and oxidant/antioxidant activity of silver and/or tellurite toward *Caenorhabditis elegans*.** Given the excellent antimicrobial activity of the silver-tellurite synergistic combination, we sought to determine whether these

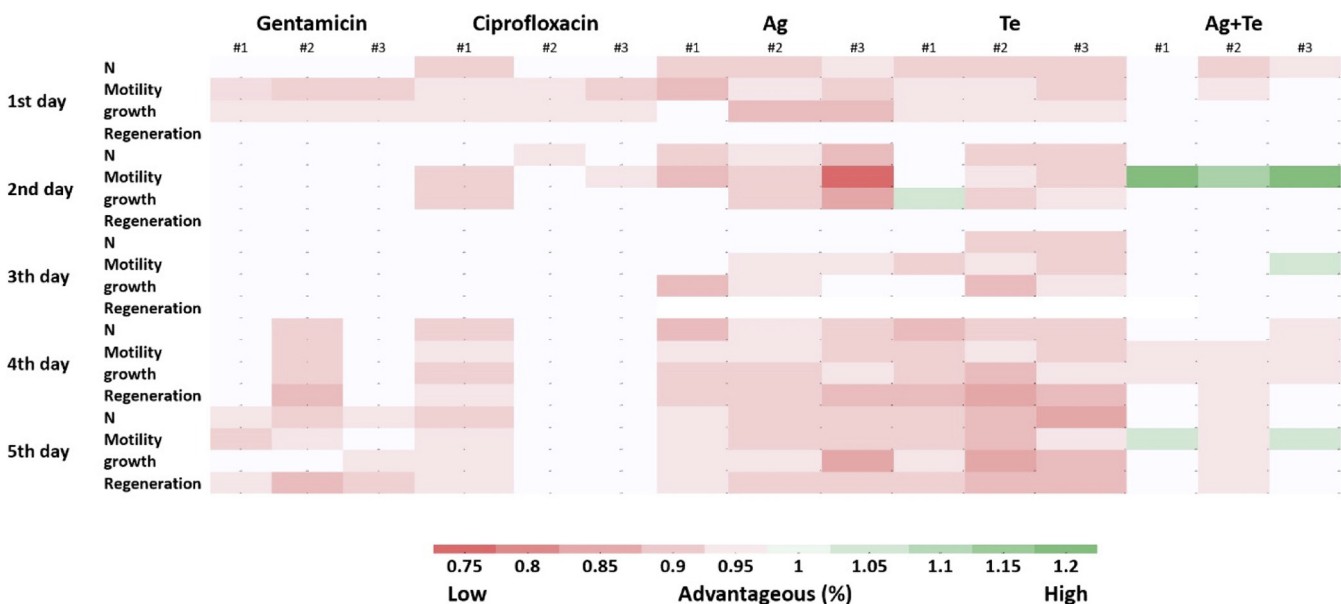

**FIG 8** Heat map on beneficial effects versus toxicity of the antimicrobials to *C. elegans*. All groups treated with silver nitrate (Ag) (1.25 mM), potassium tellurite (Te) (2.5 mM), or the Ag-Te combination (1.25 mM Ag + 2.5 mM Te) exposure, along with exposure to the antibiotics gentamicin and ciprofloxacin at 12.5 $\mu$M, were compared, and ratios were determined to the PBS-treated group. The results are reported as a ratio (%). The red color shows a harmful influence, and the groups with the green color had beneficial outcomes for the organism in comparison to the control group (total, $n = 15$; five larvae × three biological repeats). Ag, silver nitrate; Te, potassium tellurite; N, number of surviving animals.

materials would have detrimental effects on a higher eukaryotic organism. With this in mind, we chose *C. elegans* since it is a useful research model organism but also useful in toxicity testing (73) because it allows several physiological parameters to be evaluated easily and is amenable to microscopy using a variety of fluorescent probes.

Videos (see the *C. elegans* videos in the supplemental material) show the *C. elegans* advantageous features of metal(loids) in comparison to ciprofloxacin and gentamicin. The silver (see Video S1)- and tellurite (see Video S1)-exposed groups had almost the same beneficial/toxicity activity, while clear differences were found in the silver-tellurite combination group (see Video S3) compared to the antibiotic control. The silver-tellurite exposure led to even more growth, motility, regeneration (generation of offspring), and population numbers compared to the silver, tellurite, and gentamicin control (see Video S4), suggesting that the silver-tellurite combination stimulates the health of the animal. All data were compared to the control groups (see Video S5), and the results were reported as a ratio to define how various parameters lead to an advantageous (%) from the addition of the compound (Fig. 8). The heat map (Fig. 8) gives an easy visual of the data variability between biological trials and the differences in traits from a high metal challenge (10× bacterial MIC).

Reduced glutathione (GSH) plays a critical role in protecting against oxidative stress in cells, inflammation, and injury (74, 75). Here, the cellular GSH level of *C. elegans* was measured after exposure to metal(loids) and antibiotics. As shown in Fig. 9, the fluorescence intensity of silver-tellurite (694 ± 57.2 RFU [relative fluorescence units]) was twice that of silver (335 ± 84 RFU) or tellurite (302 ± 78 RFU). The remaining values were as follows: untreated control, 759 ± 65 RFU; ciprofloxacin, 722 ± 61.6 RFU; and gentamicin, 659 ± 80 RFU. This suggests stimulation of the cellular GSH antioxidant processes is strongly induced under silver-tellurite combination therapy. Direct detection of ROS levels in *C. elegans* after exposure to the metal (loids) showed that ROS (see Fig. S8) and $O_2^{\cdot-}$ (see Fig. S9) levels were also significantly lower in the silver-tellurite group in comparison to silver- and tellurite-exposed samples (for a more detailed description, see the supplemental material).

## DISCUSSION

AMR is a growing concern and leads to 1 to 5 million deaths annually (3). Therefore, there is an urgent need for novel antimicrobials. Our research team has been exploring

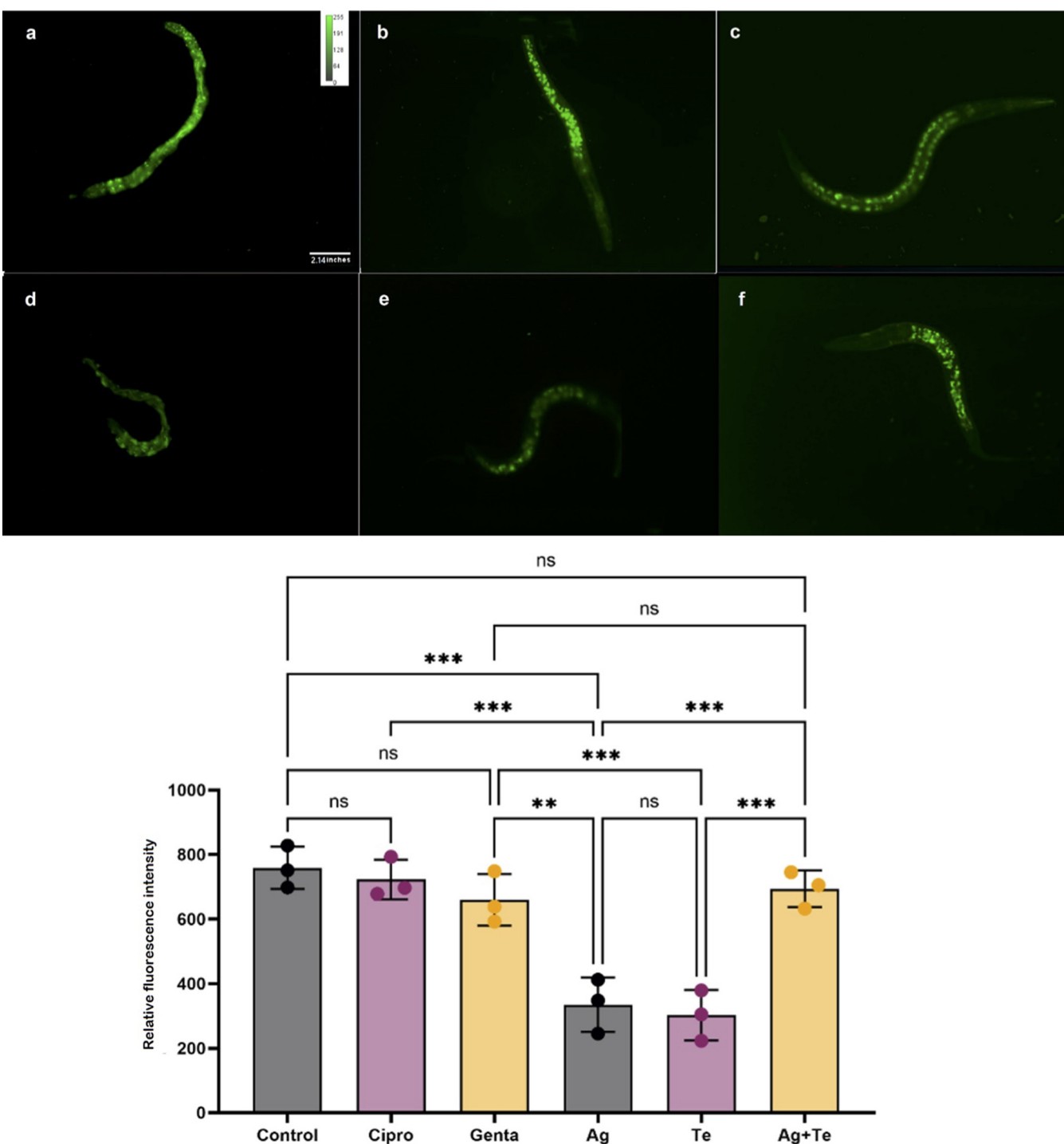

**FIG 9** Assay for reduced glutathione levels in *C. elegans* animals after exposure to antimicrobials. The fluorescence intensity of the NDA probe reflects GSH levels in *C. elegans* animals after exposure to control (PBS) (a), ciprofloxacin (12.5 $\mu$M) (b), gentamicin (12.5 $\mu$M) (c), silver nitrate (Ag) (1.25 mM) (d), potassium tellurite (Te) (2.5 mM) (e), and the Ag-Te combination (1.25 mM Ag + 2.5 mM Te) (f) (*n* = 3). The statistical differences between treatments were calculated by one-way ANOVA with *post hoc* Tukey correction (*, $P < 0.05$; **, $P < 0.01$; ***, $P < 0.001$; ****, $P < 0.0001$).

metal-microbe interactions over the last couple of decades. Metal(loids) have been used as antimicrobial agents since ancient times (76). Currently, there is renewed attention in the use of metal(loids) administered via different delivery forms as promising antimicrobial agents (13, 77, 78). Our previous studies demonstrated promising antibacterial synergistic activity of silver-tellurite against bacteria grown as either biofilm (8) and planktonic (7) forms out of thousands of different MBA combinations. The present study shows that a silver-

tellurite coapplication is quite effective against *P. aeruginosa* clinical isolates in very low concentrations compared to single metal(loid) applications. This metal(loid) duo is also superior to the commonly available antibiotics used. For instance, in most clinical isolates, silver was bacteriostatic at 1.25 mM, whereas in combination with tellurite, 0.039 mM silver had a similar result. Similarly, a 0.125 mM tellurite was required compared to when combined with silver, only 0.016 mM tellurite was required to give the same effect. Similar results were observed for bactericidal effects as well. Not only does the silver-tellurite combination have better antibacterial activity than the common antibiotics, but we have also previously shown (8) that the probability of inducing resistance toward the silver-tellurite duo is very low, and it prevents bacterial recovery as well (7). We conducted this study here to elucidate some of the mechanisms involved in the antibacterial synergism activity of the silver-tellurite combination.

Tellurite ($TeO_3^{2-}$/$HTeO_3^-$) is the prevalent form in water and the most antimicrobial form of tellurium (Te) in natural environments (79). Despite the well-known antibacterial activity of tellurite at low concentrations (7, 8, 79), and although there have been numerous studies on the mechanism(s) of toxicity of tellurite (15–21, 23, 24, 26), there are still disagreements about the specifics of the proposed mechanisms (21, 23). Further, very limited data are available on the molecular mechanism of synergism activity of tellurite. In this respect, Molina-Quiroz et al. (80) in 2012 reported that tellurite, when used in a sublethal concentration, increases the antibacterial activity of most antibiotics (10). The same research group reported tellurite increasing the antibacterial activity of cefotaxime by generating global transcriptional changes on different stress response pathways, [Fe-S] cluster assembly, protein folding, transport, and different oxidative stress regulators. It was concluded that cefotaxime with tellurite target bacteria by generating hydroxyl radical and superoxide, respectively, which leads to damaging proteins and DNA and finally bacterial death (81).

Silver is another well-known MBA and has been used as an antimicrobial since antiquity (76). Different formulations and structures of silver, especially silver nitrate ($AgNO_3$), have shown excellent antibacterial potency (11). However, increasing silver resistance in clinical isolates (7, 81, 82) and the toxicity of silver in a high concentration to humans (83) are becoming serious concerns around its use. Coapplication of silver with antibacterial agents with synergism activity would be an appropriate approach to reduce the effective concentration of silver and address resistance problems in medical applications. Several studies have shown synergistic antibacterial activity of silver with other antibiotics (11, 31, 33, 34, 84, 85), chitosan (86), probiotics (87), plant extractions (88) such as curcumin (30, 89), and other metals (7, 90) such as zinc oxide (91). Wang et al. in 2019 showed that silver mainly damages Krebs cycle by collapsing different enzymes, leading to disruption of the oxidative branch, suppressing the adaptive glyoxylate pathway, and subsequent systemic damage and bacterial death by interrupting the oxidative stress responses (29).

Unfortunately, ideas for the molecular mechanism of compounds' synergy with silver have only been suggested in a few studies (30, 34). Coapplication of silver with curcumin led to concentrating silver on the surface of the bacteria and binding of Ag to bacterial membrane. This was considered responsible for triggering high levels of ROS and membrane leakage, leading to bacterial death (30). Similarly, silver nanoparticles in a complex with synergistic antibiotics were considered to cause a temporal high release of $Ag^+$ ions near the bacterial cell wall (31). Silver combination with ebselen led to synergistic effects that inhibited thioredoxin, accelerated glutathione depletion, and increased ROS production (32). Collins et al. showed that silver can enhance the efficacy of antibiotics by oxidizing thiols, disrupting metabolism and iron homeostasis, ROS production, and membrane leakage (11). Synergistic antibacterial activity of silver with antibiotics might be because of the thioredoxin oxidative protection system disruption that leads to unabated ROS production in Gram-negative bacteria (33). However, another study reported that ATP-associated metabolism is the main reason for the antibacterial activity of silver rather than membrane leakage (34). So, although most reports suggest ROS in combination with membrane disruption, observations/conclusions are not consistent since ROS can be a function of disruption of oxidative maintenance in the cell.

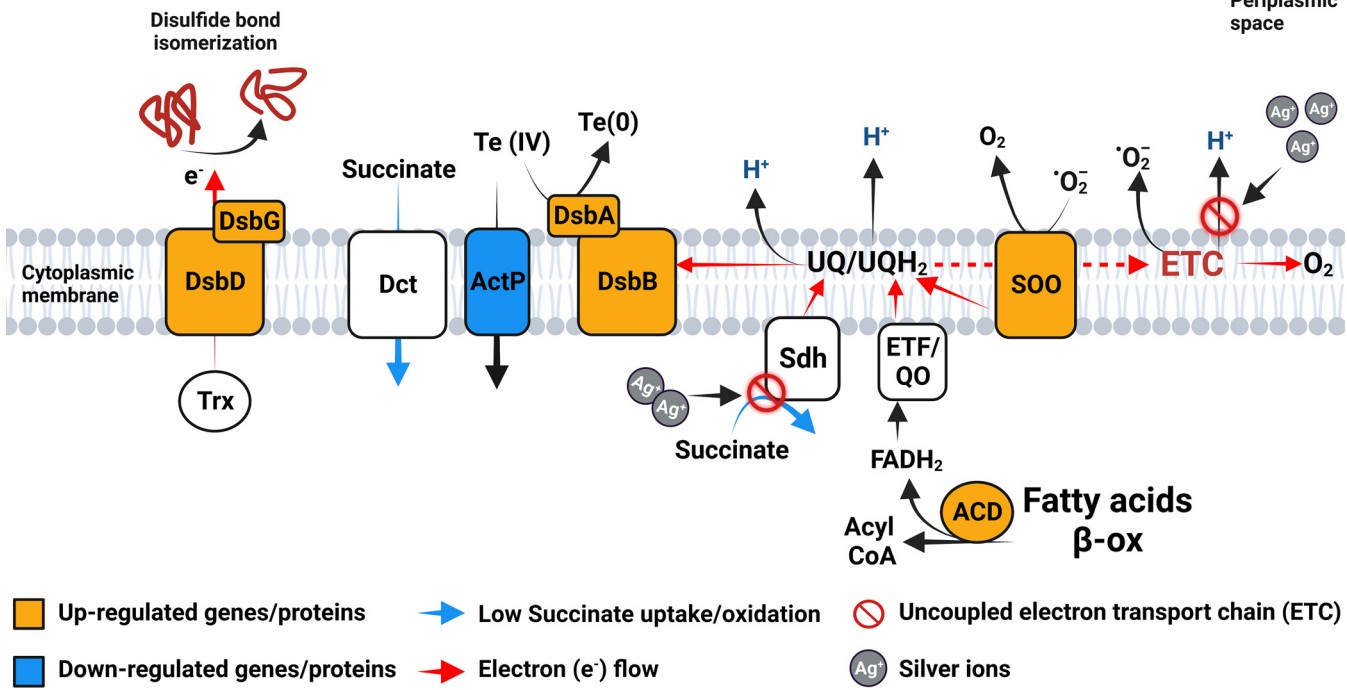

**FIG 10** Summary of key metabolic steps predicted as positively and negatively affected in cells of *P. aeruginosa* treated with the silver-tellurite combination challenge. Up- and downregulated genes and proteins are highlighted in orange and blue, respectively. In white are genes and proteins that are not differentially expressed. Red arrows highlight the electron (e⁻) flow. The red circles with diagonal lines indicate an uncoupled electron transport chain (ETC). Blue arrows indicate metabolic reactions negatively affected by the Ag-Te combination, as predicted from the metabolomic data. Te(IV), tellurite; Te(0), elemental metal tellurium; UQ/UQH₂, oxidized and reduced ubiquinone couple; ETF/QO, electron transfer flavoprotein-ubiquinone oxidoreductase; SOO, superoxide oxidase; ActP, acetate permease; Dct, dicarboxylate transporter; DsbA, -B, -D, and -G, disulfide binding proteins A, B, D, and G; Sdh, succinate dehydrogenase; Trx, thioredoxin; ACD, acetyl-CoA dehydrogenase; fatty $\beta$-ox, fatty acid beta-oxidation pathway.

Transcriptomics data provide information about gene expression profiles, which helps find pathways and systems involved in response to a change in environmental conditions. In a comparative study, we sought to find the DEG profile of *P. aeruginosa* under tellurite and silver stress, as well as their response to the synergistic antibacterial activity. Though the expression of a gene does not always lead to the gene product and functional activity for several reasons, it is still useful to obtain a system picture of response to stressors (92). Hence, we also complement transcriptomics with complementary assays of systems. The main reported mechanisms of antibacterial action for silver and tellurite were oxidative and/or ROS stress (15–17), disruption of Fe-S complex (18), cell membrane (11), and energy pathways (34).

Overall, our data agree with previous reports on silver or tellurite antibacterial mechanisms and yet provide more details. Our data showed that both silver and tellurite are targeting many pathways, but the stress is much stronger when they are applied together to the cell. We report two findings: (i) silver and tellurite target different systems, although some of these systems overlap considerably, and (ii) coapplication of silver-tellurite targets several systems in the same way as the individual metal exposure but leads to a more severe effect on those systems. This means the silver-tellurite combination does not target different mechanisms for their synergistic antimicrobial activity; rather, they simply magnify each other's effects.

Specifically, the upregulation of SOO by the silver-tellurite duo is particularly interesting in the light of the concomitant upregulation of *dsb/ccm* genes and the capacity of silver ions to interact with the components of the electron transport chain (ETC) (Fig. 10). Previous work has shown that the DsbAB/Ccm system has been directly associated with two tellurite-induced effects, namely, (i) a decrease in the reducing power of the redox chain due to oxidation of the reduced-ubiquinone (UQH₂) pool via the membrane-bound DsbA/DsbB complex (70) and (ii) an inhibition in *c*-type cytochrome assembly due to a malfunction of the Ccm system (93). In regard to the silver-induced effects, Ag⁺ enhance the release ROS

at the level of the ETC by physically interacting with the cytochrome *bd* terminal oxidase and succinate dehydrogenase (Sdh) (29, 94). However, these negative effects are possibly counterbalanced by the SOO activity, whose function is to oxidize superoxide anions with the consequent reduction of the ubiquinone pool (93). In addition, if SOO reduces the quinone pool by taking protons from the cytosol (93), the DsbA/DsbB complex reoxidizes the $UQH_2$ pool on the periplasmic side (70), thus generating an electrochemical potential gradient (PMF). Therefore, the net loss of membrane potential caused by the production of superoxide anions through electron leakage, mostly at the level of the uncoupled ETC, is possibly compensated by the electrochemical gradient linked to SOO activity (Fig. 10). Based on this proposal, the mechanism triggered by tellurite is seen as a turbo system since it recycles a waste/unwanted product ($O_2^{--}$), limiting the overall losses of the process. The latter effect, although it may seem an apparent paradox, is only one of the many phenomena that characterize the overall picture of cellular toxicity induced by metals and metalloids at the plasma membrane level. In this regard, the decrease of succinate oxidation in cells treated with silver-tellurite is interesting, and it is probably attributable to two factors: (i) a restriction in the consumption of succinate at the level of the quinone pool and (ii) a decrease in the activity of the dicarboxylic acid transporter (Dct) whose function depends on the proton gradient (48) (Fig. 10). In fact, as reported in "Release of iron from proteins" above, the toxic effects of silver-tellurite are attributable, among the many observed, to a membrane uncoupling, while the upregulation of SOO and of acyl-CoA dehydrogenase (ACD; A4W92_RSO03220; see Table S3), which generates reducing agents for the ETF:QO (Electron Transferring Flavoprotein:Quinone Oxidoreductase), would lead to an overreduction of the ubiquinone pool. Since it is known that the maximum rate of input and output of electrons from the quinone pool occurs when the pool is predominantly oxidized (70), it is likely that in cells treated with silver-tellurite not only is less succinate transported into the cytosol but also the rate of that succinate oxidation is slowed down compared to control cells.

The oxidizing/toxic effect of tellurite at the level of the periplasmic space is also evident from the upregulation of two Dsb proteins which are part of the Ccm system, namely, DsbD and DsbG. These two disulfide-binding proteins maintain the periplasm in a partially reduced state by transferring reducing agents from cytosolic thioredoxins (Fig. 10) (95). As a result, the upregulation of *dsbDG* leads to an imbalance of the redox state of the cytosol, as already evidenced in the past by the decrease of the reduced glutathione (GSH) pool in the presence of tellurite (95).

Strictly related to this is the effect on other membrane proteins, such as the downregulation of the acetate permease (ActP, A4W92_RS25380; see Table S3) by exposure to the silver-tellurite duo. It has been shown in the past that tellurite oxyanions are nonspecifically transported into cells via multiple pathways, including acetate permeases (ActP1 and ActP2), using the $\Delta$pH component of the electrochemical gradient (21). We observe that the *actP1* gene of *P. aeruginosa* is upregulated by silver but strongly downregulated by the silver-tellurite combination. Thus, the copresence of tellurite generates a sort of rescue effect which represses its entry via ActP1. On the other hand, keeping tellurite out of the cells only accelerates the oxidation mechanisms of the periplasmic space described above with an evident short-circuit in the entire cell redox balance (21, 93).

Recently, other studies of mixed metal stress have been explored using different 'omics approaches against different organisms to appreciate responses of multi-metal toxicology. In a case of aluminum and indium toxicity to *Daphnia magna*, a potential synergy of more than an additive response in shared systems and enhanced response of 40 shared genes was observed (96). However, here in this example, high numbers of unique genes were seen under the dual challenge. A previous example of evaluating additive effects was the study of the response of *Aspergillus fumigatus* under stress from six metals (lead, copper, nickel, cadmium, zinc, and chromate) at the same time. In that study, cytochrome oxidase was found to be the major player, reflecting in this fungus a response similar to bacteria that the electron transfer chain is primary player (97). A study using an environmental isolate of *Bacillus cereus* from a metal-polluted site upped the challenge to eight different metals (aluminum, uranium, manganese, nickel, cobalt, copper, iron, and cadmium) and found that the primary

response was disrupted iron homeostasis leading to the collapse of cell energetics (98). Within the metal toxicity field, there have been studies of metal exposure to many different life forms, and the data tend to be complex, with surprises being unmasked in mixed-metal exposure studies. We noted in our earlier study evaluating the physiochemical characteristics of the metals to toxicity to bacteria that there was no correlation to ionization potential or the hydrolysis potential and yet these are correlated well with metal toxicity to higher organisms (99). In the present study, we see that the metal combination is toxic to bacteria but helpful to the worm. Perhaps we are seeing something to explore more deeply, where there is a common thread with microbes in the disruption to bioenergetics and less so in higher organisms.

Apparently, the image that emerges from the multiexperimental approach we describe here is that of a bacterial cell in a physiological state of suffering preceding its inevitable death. In analogy with the definition used to describe the state of organic decay in higher eukaryotic organisms, the prevalence of the negative/toxic effects induced by the silver-tellurite duo can be conceived as a state of cellular cachexia. In this respect, it is particularly noteworthy that the animal model used here, *C. elegans*, showed that under either silver or tellurite challenge (in $10\times$ the MIC) has slightly higher toxicity in comparison to ciprofloxacin and gentamicin. However, the coapplication of silver-tellurite reduces their toxicity to the host drastically. Moreover, the coapplication of the metals led to increased antioxidant properties of the host and, as such, the silver-tellurite exposure to bacteria is highly antimicrobial, and the worms are healthier. Our previous meta-analysis review suggests that tellurite has less toxicity to human cells in comparison to silver (8). The U.S. Food and Drug Administration approved the use of silver as an antibacterial for treating topical infections at concentrations of $\sim$6 mM (100). This is a considerably higher load than what our study suggests for clinical isolates, which displayed an MIC range of 0.62 to 2.5 mM. Moreover, the synergistic combination decreased their effective antibacterial concentration drastically, which would allow for a far lower metal(loid) exposure to the host and, as such, we see the benefits of this metal-metalloid powerful combination.

## MATERIALS AND METHODS

**Bacterial strain, culture media, stock, and working metal(loid)-based antibiotic solutions.** *P. aeruginosa* strains ATCC 27853 and PAO1 and 39 clinical isolates from the Alberta Health Services Regions were used in this study. Detailed information on the culture media and metal(loids) is provided in the supplemental material.

**Synergism and bactericidal and bacteriostatic efficacy of silver-tellurite combination against clinical isolates in simulated wound fluid.** In our previous study, the bacteriostatic and bactericidal synergism activities of 5,760 combinations of MBAs were screened against reference strains in lab Luria-Bertani medium, Muller-Hinton broth, and simulated wound fluid (SWF). After initial screening, the silver nitrate (AgNO$_3$; abbreviated to elemental symbol "Ag")/potassium tellurite (K$_2$TeO$_3$, abbreviated to "Te") combination was selected as the most effective antibacterial combination (7). In another study, the potential of synergistic biofilm prevention and eradication potentials were screened systematically in a total of 1,920 combinatorial MBAs. Similarly, the silver-tellurite combination was identified as the most effective agent against *P. aeruginosa* biofilm (8).

Here, the antibacterial activity of the silver-tellurite combination was explored against clinical isolates. The silver-tellurite combination was tested on 39 clinical isolates. Both the bactericidal and bacteriostatic potency of Ag and Te, as well as their synergism effects, were tested in SWF as a surrogate of wound environment and compared to gentamicin and ciprofloxacin as the most common antibiotics used against *P. aeruginosa* infections (101, 102).

The methodology for determining the MIC for bacteriostatic activity and the MBC for bactericidal potency measured by the recovery potency of bacteria after exposure with each MBAs is provided in the supplemental material.

**Sample preparation for RNA-seq and metabolomics.** *P. aeruginosa* ATCC 27853 was cultured in 3 mL of SWF and incubated at 37°C in a shaker incubator (150 rpm) to reach an optical density at 600 nm (OD$_{600}$) of 0.8. The culture was then challenged at the MICs of the agents (0.125 mM silver nitrate [Ag], 0.25 mM potassium tellurite [Te], or 0.125 mM Ag/0.25 mM Te combination), control groups (treated with phosphate-buffered saline [PBS]), and incubated at 37°C for 2 h in a shaker incubator (150 rpm). The example growth curve in Fig. S9 reflects that over the time frame of the metal challenge, the bacteria became stressed but did not stop growing and that the cell harvest occurred before loss of cell significant cell density would be a concern in the comparison of data to the unchallenged controls. Samples (1 mL) were centrifuged at 3,500 $\times$ $g$ for 10 min to pellet the cells. The cell density here was normalized to 1.5 $\times$ 10$^8$ CFU/mL bacteria for both challenged and unchallenged samples. The pellets were extracted and used for RNA sequencing (RNA-seq), and the supernatant was collected for the metabolomics study. Three biological repeats on three different days were used for each sample. The total samples ($n = 12$) used for RNA-seq were as follows: silver ($n = 3$),

tellurite ($n = 3$), silver-tellurite ($n = 3$), and control (treated with PBS; $n = 3$) (see Table S2). A total of 24 samples were used for metabolomics because three different controls without bacteria (metalloids + media) were considered for each data set, one from each biological trial. This control was in place in case the metals catalyzed any chemical changes to the medium components on their own.

**RNA isolation and rRNA depletion.** After treatment, zirconia beads were used for cell lysis for at least 2 min using a bead beater instrument (FastPrep-24; MP Biomedicals, Solon, OH). The total RNA was then extracted with a RiboPure Bacteria RNA isolation kit (Invitrogen, USA) according to the manufacturer's recommendations. To ensure that the DNA was completely removed, residual DNA was removed with DNase (Sigma-Aldrich, Inc, St. Louis, MO) to get a pure RNA sample. RNA quality and quantity were analyzed using a spectrophotometer (NanoDrop Technologies; Thermo Fisher Scientific, Inc., Wilmington, DE). Samples were prepared with a New England Biolabs NEBNext rRNA depletion bacteria kit and a NEBNext Ultra II Directional RNA library prep kit (Illumina) according to the manufacturer.

**Transcriptomics and RNA-seq.** Samples were sent to the Centre for Health Genomics and Informatics and UCDNA Sequencing University of Calgary Cumming School of Medicine for sequencing. Table S2 shows the sample information and summary of cDNA samples for RNA-seq. After adaptor ligation and amplification, the average library sizes were 323 to 344 bp. The libraries were quantified using a KAPA qPCR Library Quantification kit for Illumina platforms and then pooled and loaded onto the Illumina NextSeq500 sequencer at 1.8 pM, according to Illumina guidelines (103, 104). The libraries were sequenced using a 75-cycle high-output sequencing kit.

**Quantification of gene expression and differential expression analysis.** Quantification of gene expression and identification of DEGs from RNA-seq was performed by using Salmon in combination with DESeq2. For quantification of gene expression, RNA-seq reads were mapped using Salmon v1.9.0 (105) against the *P. aeruginosa* ATCC 27853 transcripts downloaded from NCBI RefSeq (GCF_001618925.1). After mapping, Salmon "quant" files were used to generate a gene-level expression matrix with tximport v1.22.0 (106). The expression matrix was finally used for the analysis of DEGs with DESeq2 v1.34.0 (107) by setting the $P$ value cutoff for multiple testing adjustment to 0.01. Genes were selected as differentially expressed based on a false discovery rate (Benjamini-Hochberg correction) of $<0.05$ and a |$\log_2$-fold change| of $>1.0$. Gene Ontology (GO) terms were assigned to ATCC 27853 genes using eggNOG (108), and the identification of significantly overrepresented GO terms in *P. aeruginosa* DEGs was performed using topGO v2.46.0 (https://bioconductor.org/packages/release/bioc/html/topGO.html). *P. aeruginosa* protein-coding genes were further annotated through the KEGG database using KEGG KofamKOALA (109). The complete DESeq2 results along with KEGG annotation and topGO enrichment analysis are available as Tables S3 to S5 in the supplemental material.

**Metabolomic workflow.** Samples for metabolomics were taken from the same cultures as the transcriptomic experiments. The cultures cell densities were determined to be $1.5 \times 10^8$ CFU/mL for all samples. A 10-$\mu$L aliquot of the supernatant was obtained by centrifuging each sample tube from the cell culture used in the transcriptomics study, providing the extracellular metabolites within this spent media; thus, all of the conditions for transcriptomics and metabolomics were identical. Each cell-free aliquot was added to a well in a 96-well PCR plate (VWR 96-well real-time PCR skirted plate), to which 90 $\mu$L of 50% methanol (Fisher Optima) and 50% water (Fisher Optima LC/MS grade) was added. The plate was centrifuged at 4,000 rpm for 10 min (Thermo Sorvall Legend XTR). Subsequently, 70 $\mu$L of the supernatant was transferred to a new 96-well sampling plate (Masterblock; 0.5 mL V-bottom, sterile; Greiner Bio-One, Monroe, NC) and diluted with 70 $\mu$L of 50% methanol for a total sample dilution of 1:20 from the starting concentration. This yielded extracellular metabolites for analysis (110).

The samples were analyzed using a Thermo Scientific Q Exactive HF Hybrid Quadrupole-Orbitrap mass spectrometer (UHPLC-MS; Thermo Scientific). Mass data were collected in negative-ion mode using a full scan from 50 to 750 $m/z$ at a 240,000 resolution, with automatic gain control (AGC) target of $3e^6$ and a maximum injection time of 200 ms. MAVEN (El-MAVEN v0.12.0) software was utilized to analyze the acquired data and for supervised peak picking (111). The cell density of the samples prior to centrifugation was found to be $1.5 \times 10^8$ CFU/mL bacteria for all samples, indicating that the chosen Ag and Te concentrations did not reduce growth during the 2-h exposure (although metal-exposed cultures were found to be nonviable after 8 h). Thus, no normalization was required for adjustment in cell densities, and the results would therefore be due to differences in physiology between the various exposed cultures.

**Hydrogen peroxide assay.** The hydrogen peroxide concentration after exposure with metal(loids), ciprofloxacin, and gentamicin was detected by using a Pierce quantitative peroxide assay kit in an aqueous compatible formulation according to the manufacturer's instructions (112). To prepare the standard, a 1 mM solution of $H_2O_2$ was initially made by diluting a 30% $H_2O_2$ stock to 1:9,000 (11 $\mu$L of 30% $H_2O_2$ into 100 mL of double-distilled water [DDW]). This sample was then serially diluted with DDW to 1:2 (100 $\mu$L of DDW + 100 $\mu$L of the previous dilution) for a total of 11 samples as a standard. Then, 200 $\mu$L of the working reagent from the kit was added to 20 $\mu$L of the diluted $H_2O_2$ standards. Samples were mixed and incubated for 15 min at 21°C in the dark. Absorbances were measured at 595 nm using a Thermomax microtiter plate reader with Softmax Pro data analysis software (Molecular Devices, Sunnyvale, CA).

To measure the treated and untreated samples, overnight incubations were subcultured in 3 mL of SWF and then incubated at 37°C in a shaker incubator (150 rpm) to reach an OD$_{600}$ of 0.8 ($1.5 \times 10^8$ CFU/mL). They were then treated with MICs of agents (0.125 mM silver nitrate [Ag], 0.25 mM potassium tellurite [Te], or 0.125 mM Ag + 0.25 mM Te [combination]) and other groups were treated only with PBS (as a negative control). Ciprofloxacin and gentamicin at 1.25 $\mu$M were used as antibiotic comparators. A positive control for the assay was treated with 250 $\mu$M $H_2O_2$, and all samples were incubated at 37°C for 2 h in a shaker incubator (150 rpm). Little change in cell density occurred over this time frame since the MIC levels did not slow growth or division significantly over this time frame (see above). The bacterial cells were washed with PBS by centrifuging

(10,000 rpm for 5 min) and discarding the supernatant. Next, 3 mL of PBS was added to each sample, following by vortexing. Then, 200 $\mu$L of the working reagent was added to 20 $\mu$L of each sample. Samples were mixed and then incubated for 15 min at room temperature. Absorbances were measured at 595 nm using a Thermomax microtiter plate reader with Softmax Pro data analysis software (Molecular Devices). The value of a PBS blank was subtracted from all sample measurements. $H_2O_2$ concentrations were calculated by using a standard curve of defined $H_2O_2$ concentrations.

**Membrane permeability measurements.** For the measurement of membrane permeability, propidium iodide (PI; Invitrogen, Eugene, OR) was used as the fluorescent reporter dye. Increased PI red fluorescence is correlated with increased membrane disruption and permeability since it can enter and remain in the cells and bind to DNA (11, 112). The bacteria were cultured in 3 mL of SWF and incubated at 37°C in a shaker incubator (150 rpm) to reach an $OD_{600}$ of 0.8 as in all experiments. Each of the groups was treated with MICs of agents or PBS (as a negative control), and the bacteria were boiled at 90°C for 10 min (as a positive control); all samples were incubated at 37°C for 2 h in a shaker incubator (150 rpm). The samples were centrifuged and washed with PBS (10,000 rpm for 2 min). The cells were then incubated with 0.16 mM PI for 5 min at 21°C in dark, and then 10-$\mu$L portions of the samples were transferred onto slides and examined on a fluorescence microscope (Zeiss Axio Imager Z1) at the same exposure time (640 ms). Color densitometry analysis was performed by using Fiji software (ImageJ).

**Advantageous/harmful and oxidant/antioxidant surveys in *Caenorhabditis elegans* as an animal model.** For the *C. elegans* assay, the detailed method is provided in the methods section of the supplemental material. In all, four to six synchronized L2 larvae were transferred to agar plates containing 10× the MIC of each antibacterial agent. Under this antimicrobial load, the food for the worms (*E. coli* bacterium cells) will have died (but do not lyse) during the first day of the experiment and, by the end of the experiment, the worms would have run out of food, thus the offspring would not mature. This did not affect our observations over the timescale of our experiment. Four results were recorded for 5 days: (i) number of live worms, (ii) motility (head swing and body-bending frequency per minute), (iii) growth (body length and body width), and (iv) regeneration (generation of offspring, size, number, and motility of them were considered) rates.

Multiple fluorescent sensor probes were employed, including dihydroethidium (DHE), 2′,7′dichlorofluorescin diacetate (DCFH-DA), and naphthalene-2,3-dicarboxal-dehyde (NDA), all obtained from the Invitrogen to detect the $O_2^{\cdot-}$, ROS, and glutathione, respectively.

**Statistical analysis.** GraphPad Prism 9 software (113) was used to calculate the statistical differences between different treatments (three technical and biological replicates for each treatment group). One-way analysis of variance (ANOVA) with a *post hoc* Tukey correction was used, and significance is indicated by asterisks in the figures (*, $P < 0.05$; **, $P < 0.01$; ***, $P < 0.001$; ****, $P < 0.000$). In other cases, the significance between the two groups was determined by a two-tailed Student $t$ test. All results are expressed as means $\pm$ the standard deviations. All experiments were repeated with at least three biological replicates.

**Data availability.** Transcriptomic data are available under NCBI GEO accession number GSE232832. Additional data are available in the main text or the supplemental materials.

## SUPPLEMENTAL MATERIAL

Supplemental material is available online only.
**SUPPLEMENTAL FILE 1**, DOCX file, 4.3 MB.
**SUPPLEMENTAL FILE 2**, XLSX file, 2.5 MB.
**SUPPLEMENTAL FILE 3**, DOCX file, 0.01 MB.
**SUPPLEMENTAL FILE 4**, MP4 file, 11.9 MB.
**SUPPLEMENTAL FILE 5**, MP4 file, 6 MB.
**SUPPLEMENTAL FILE 6**, MP4 file, 10.9 MB.
**SUPPLEMENTAL FILE 7**, MP4 file, 3.9 MB.
**SUPPLEMENTAL FILE 8**, MP4 file, 3.3 MB.

## ACKNOWLEDGMENTS

Funding to R.J.T. and D.H. was provided by individual discovery grants from the Natural Sciences and Engineering Research Council of Canada. A.P. was supported by a MITACS Elevate Postdoctoral Fellowship in partnership with C-Crest Laboratories, Inc., Montreal, Canada. A.F. was also partially financed by the Italian Ministry of Education, Universities, and Research through the REACT EU program. M.M. was supported by a Mitacs Accelerate Postdoctoral Fellowship. All metabolomics was carried out in the Calgary Metabolomics Research Facility.

Author contributions were as follows: conceived and designed the study—R.J.T. and A.P.; practical performance—A.P., A.F., M.M., and D.A.S.-A.; analyzed the data—A.F., D.A.S.-A., M.M., M.C., R.J.T., and A.P.; wrote the first draft of the paper—R.J.T., A.P., A.F., M.Z., and D.Z.; and participated in data analysis and manuscript editing—A.P., A.F., D.A.S.-A., M.M., D.H., M.C.,

D.Z., M.Z., and R.J.T. All authors have read and agreed to the published version of the manuscript.

We declare there are no competing financial interests.

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
