## [Reviewer comments · Microbiology Spectrum]

Microbiology Spectrum

Insights into the Synergistic Antibacterial Activity of Silver Nitrate with Potassium Tellurite Against *Pseudomonas aeruginosa*

Ali Pormohammad, Andrea Firrincieli, Daniel A. Salazar- Alemán, Mehdi Mohammadi, Dave Hansen, Martina Cappelletti, Davide Zannoni, Mohammad Zarei, and Raymond Turner

Corresponding Author(s): Raymond Turner, University of Calgary Department of Biological Sciences

Review Timeline:

Submission Date:	February 11, 2023
Editorial Decision:	May 3, 2023
Revision Received:	May 26, 2023
Editorial Decision:	May 31, 2023
Revision Received:	June 3, 2023
Accepted:	June 5, 2023

Editor: Paolo Visca

Reviewer(s): Disclosure of reviewer identity is with reference to reviewer comments included in decision letter(s). The following individuals involved in review of your submission have agreed to reveal their identity: Jennifer L Goff (Reviewer #3)

Transaction Report:

DOI: <https://doi.org/10.1128/spectrum.00628-23>

May 3, 2023

Dr. Ali Pormohammad
University of Calgary
24 Ave NW
Calgary
Canada

Re: Spectrum00628-23 (Insights into the Synergistic Antibacterial Activity of Silver Nitrate with Potassium Tellurite Against *Pseudomonas aeruginosa*)

Dear Dr. Ali Pormohammad:

thank you for submitting your manuscript to Microbiology Spectrum.

Your manuscript has now been reviewed by three experts in the field who provided their constructive criticism by highlighting several inconsistencies or weaknesses that need careful revision. I agree with the reviewers' comments and I believe that an in-depth revision of your manuscript, both conceptual and stylistic, is needed to reach the quality standard required by Microbiology Spectrum. Therefore, I am asking you to extensively revise your manuscript by addressing, one by one, all reviewers comments.

Link Not Available

Sincerely,

Paolo Visca

Journals Department
Reviewer comments:

Reviewer #1 (Public repository details (Required)):

RNA-seq data needs to be uploaded to GEO

Reviewer #1 (Comments for the Author):

In this work Pormohammad, A et. al used transcriptional, metabolomics and biochemical assays to understand the toxicity of silver and tellurite when administered together as a strategy to kill *Pseudomonas aeruginosa*. The joint administration of these metals is more toxic for bacteria than the exposure to metals alone in laboratory strains and clinical isolates. Global transcriptional changes are reported which seem to be correlated with metabolomic and biochemical data. Even when this work is interesting due to the need to improve the current antibacterial therapies, there are many aspects that need to be improved as will be mentioned.

Minor comments

The method to determine MBC from cultures previously used to assess MIC might not be very accurate since a 1:15 dilution is not high enough to get rid of the antibiotic in cultures exposed to higher concentrations of drugs (which may inhibit bacterial growth). Since antibacterials were not washed away and authors do not mention the range and dilutions (related to MIC) of concentrations used it is not evident to determine if the values of MBC are correct. A more accurate method to determine MBC is to determine the number of CFU counts after treatment with different concentrations of antibacterials.

The way to start cultures needs to be specified. It is not evident if the 1.0 McFarland culture was generated from an overnight culture or from colonies scrapped from a plate.

In figure 1 it would make much easier for reader to analyze data if the average of biological replicates (repeats) were shown into only one colored square/number.

RNA-seq data needs to be uploaded to GEO and an accession number has to be included in this manuscript.

All the experiments in this manuscript were conducted by exposure of cultures to metals alone or in combination for 2 hours, except for those showing the effect of antimicrobial of membrane integrity (Figure 7). Authors need to explain why they did that, if they want to compare with their other results, they need to be consistent.

It is not evident for this reviewer the point that authors want to state with the experiments using *C. elegans*. A better description may improve the understanding of these results and why they need to be included in this manuscript.

Major concerns

The transcriptional profile experiment compares exponentially growing cultures treated with Ag, Te or Te+ Ag, with a PBS treated culture that keep growing and will eventually reach stationary phase (implying a completely different metabolic and transcriptional status). Cultures treated with metals alone or in combination (at MIC) will stop growing. It is not accurate to say that bacteria "is responding" to a treatment if the initial condition is not known (as mentioned throughout the whole manuscript), so a most proper control would be to compare the transcriptional status post treatment with metals to the one that bacteria display before treatment (prior to the addition of toxicants). Otherwise, language needs to be softened and modified as "global transcriptional changes" instead of cell/bacterial response. Also, since in stationary phase there is a lack of nutrients which generates a global stress situation (oxidative stress included), these results might underestimate their findings when cultures are challenged with metals. Authors should discuss this possibility in the text.

Since authors are comparing the total amount of metabolites present in cultures exposed to different antimicrobial compounds, experiments shown in Figures 5, S6, S7 need to be normalized ideally by CFU counts or by total protein content, especially since differences between experimental conditions are subtle (2-6-fold). Even at the MIC those compounds will be killing cells differently when alone or in combination, so a 2-6 difference in the total amount of cells in the assay may imply that all the conditions display no differences with each other. For this reason, it is not possible to compare cultures challenged with metal(oids) alone or in combination without proper normalization.

A good example of this is the comparison of H₂O₂ levels shown in Figure 6 when authors compare levels to negative control (untreated cultures) which after 2 hours growing will reach stationary phase containing more than 100X more cells than the treated conditions. Those differences in the total amount of cells used in the assay are not displayed in the experiments of this manuscript.

In figure 8 authors suggest that combined administration of silver-tellurite stimulates the health of the animal. However, in the experimental design they mention "The *C. elegans* wild-type strain N2 (Bristol) and *E. coli* OP50 at 10⁹ CFU/mL as a food source were used in this study (25, 26). *E. coli* and 10 times the MIC of each antibacterial component were added to the agar plates." Since *E. coli* OP50 will be facing lethal concentrations of metals is it possible that *C. elegans* prefers death or dying cells to grow? Authors should discuss this possibility.

Brighter rod-shaped structures are observed in Ag-Te treated worms in Figure S8. Are these bacteria? If so, this observation may imply that Ag-Te administration kills *E. coli* OP50 generating oxidative stress as has been reported by other groups. These findings may support the previous comment suggesting that it is easier for *C. elegans* to be fed by dead bacteria.

Reviewer #2 (Comments for the Author):

This is a very interesting work that adds to both basic scientific knowledge on the subject, and also has potential applications. I have a few recommendations for the paper:

- 1) I would recommend moving the sound from the supplementary videos and fixing the titles - while watching the videos, I could hear a recording of a conversation in the background. Some of the titles are not fully visible in the video, and there is a typographical error in at least one of them. The captions could also indicate what the arrows are pointing out.
- 2) There are a number of grammatical errors/words missing or incorrect, etc. in the manuscript, for example at lines 114-115; 293-294; 298-302; 312; 345; 355; 394; 400; 435-439; 444-445; 458; 465-467; 499; 522-523; 638; 640-642; 658; and in the captions for Figures 4, 6, and 9. There are also a number of errors in the supplementary material including a few incorrect words. There are errors in the titles for Figures S8 and S9 as well. I would recommend having someone go over the manuscript for these types of errors, as that will make it easier to read.
- 3) There is a problem with reference 98 in the main manuscript, as written, and also with references 5 and 27 in the supplementary material - I would recommend carefully reviewing the references and ensuring they have all been entered correctly.
- 4) Figures S1 and S2 have very small print. It would be better to spread them out over a few pages so that they are easier to read.
- 5) If the tellurite is just enhancing the mechanisms of action of silver treatment, in situations where silver resistance develops (i.e. to those mechanisms of action), presumably adding tellurite would not help? Perhaps the authors could address this.

Overall, these changes are quite minor, in an otherwise extensive and informative paper.

Reviewer #3 (Public repository details (Required)):

Transcriptome raw reads should be deposited in SRA or equivalent database. Submitting expression data to NCBI GEO or the EMBL equivalent would also be nice as it would enhance the extractability/usability of the dataset.

Reviewer #3 (Comments for the Author):

This manuscript describes a systems-level characterization of the synergism between tellurite and silver in their toxicity towards *Pseudomonas aeruginosa*. Studies of metal(loid) mixtures, in general, is an emerging area in the microbiology world. I believe that both clinical and environmental microbiology researchers would find interest in this paper and I am enthusiastic to see it published after revisions. The authors initially show nice synergism between the two metal(loids) in various *P. aeruginosa* clinical isolates. They then go on to characterize the transcriptome, metabolome, and other biochemical responses of a laboratory strain of *P. aeruginosa* to the combined treatment relative to the individual exposures. I really liked the integration of the physiological data at the end with the transcriptomic data. These data fit together nicely. In contrast, the metabolomics data was not well integrated at all, and I was left wondering why it was even included since it felt very "isolated" from the other data. Further work needs to be done to better integrate these data into the story. Finally, the paper needs thorough English language editing from either a fluent co-author, a trusted fluent colleague, or a professional editing service. Specific comments follow:

Major:

1. It looks like I am missing the supplemental excel tables with all the gene expression and metabolome data. Please include those in the next submission. Also, consider depositing your raw reads in the SRA (or equivalent). Gene expression data would also be more accessible if deposited in an archive like NCBI GEO.
2. This manuscript represents one of a relatively limited number studies that have begun looking at metal(loid) mixtures on a systemic level (i.e., using omics-based approaches) in microorganisms. Since this is such an emerging and exciting area of research, it may be useful to look for connections between your work and those other studies. This would be much more interesting and novel than the extended descriptions of individual tellurite and silver toxicity in the Discussion sections-topics that have been very well-reviewed many times over. For examples, are there certain emergent properties of all metal mixtures regardless of composition? Or is each mixture its own unique "thing"? In overlap in differentially expressed pathways? Even tying in some of the senior author's own prior work on metal(loid) mixture impacts on microbial physiology within the discussion would be extremely interesting/informative. Some suggested papers:

Dey, Priyadarshini, et al. "Insight into the molecular mechanisms underpinning the mycoremediation of multiple metals by proteomic technique." *Frontiers in Microbiology* 13 (2022). Obviously this first paper is on a fungus (*Aspergillus*), but I think there may actually be some overlap with the data you highlight here

Goff, Jennifer L., et al. "Mixed heavy metal stress induces global iron starvation response." *The ISME Journal* 17.3 (2023): 382-392.

Bacillus and under anaerobic conditions (so may have some significant differences due to that...hard to judge, though, since I don't have the complete transcriptome dataset)

Brun, Nadja R., et al. "Mixtures of aluminum and indium induce more than additive phenotypic and toxicogenomic responses in *Daphnia magna*." *Environmental science & technology* 53.3 (2019): 1639-1649.

This last one would be more relevant to the metal mix effects in your nematode system. And, in general, there are a lot more studies out in the multicellular eukaryotic world on systems-level metal mixture effects that could be drawn upon to add context to your nematode observations.

3. Line 128: For the uninitiated, it might be useful to state if *C. elegans* is a useful model for pre-human testing toxicology studies. Otherwise, I am left wondering, why are you studying metal toxicity in a nematode.

4. Line 132: At the beginning of your results, you need to explicitly state that all work was conducted under aerobic conditions as oxygenation can significantly impact the toxicity of metal(loid)s

5. Line 166: You need to lead this section by immediately discussing exposure conditions. So, move section 2.2.1 up top. You can include the PCA details after you introduce the exposure conditions.

6. Section 2.3: Can you better integrate the metabolomic data with the transcriptome data. You do a beautiful job of this in Section 2.4. I would love to see something similar done with this section. For example, do any DEGs correlate with metabolite changes? Is there a figure you could create of metabolic pathways to highlight the integration of the data? Although, I realize this is extracellular data so maybe that's not feasible.

7. Line 346: tellurite reduction to Te by glutathione will also release superoxide. So, there may be multiple ROS sources.... With that said, do you see evidence of tellurite reduction in your cultures?

8. Discussion paragraphs 2 and 3: as already mentioned above, these paragraphs were too long and "review-y". They really took me out of the story since you aren't tying it to any of your results. I imagine they could be easily condensed to a couple of sentences each briefly stating (1) major mechanisms of individual metal(loid) toxicity and (2) other mixtures synergism has been observed in.

9. Line 504: if Ag exposure alone is increasing actP expression, then how would there be an increase in tellurite uptake? There is simply no tellurite in the system to be taken up. Perhaps this is just a confusing sentence that needs re-wording

10. In the final couple paragraphs of the discussion you present a model to describe the synergism between the two metals. I can see where its going, but it is very hard for me to visualize in my head. Could you prepare a figure to illustrate this?

11. Figure 1: very visually busy figure that would probably be better as a table with averaged values (+/-SD). You could potentially still color the table concentration

12. Most figures: where relevant, I would like to see exposure concentrations given on figures or in the legends (but preferably on the figures themselves)

13. Figure 8: again, this is an instance where a table summarizing these data may be more helpful. You can simply average your replicate groups. Showing each individual replicate make the table too big and challenging to read. However, other possibilities include transforming into individual scatter plots and only showing the most interesting ones in the main text

14. Just a comment relating to future studies, but have you considered working with the organotellurium(IV) drug AS101? It has been used as an antimicrobial (among other things) in mammalian systems so its pharmacological profile is pretty well-characterized and may be more feasible for trials in mammalian systems (if that is your goal).

Minor:

1. There is inconsistent capitalization of chemical names (particularly tellurite, silver, and metabolites mentioned later in the text). These should be lowercase unless, of course, they are the first word of the sentence.

2. Line 80: "The Lancet" should be italicized

3. Line 160: Are these MICs/MBCs really the same for all the clinical isolates? Table 1 seems to suggest not. Please edit for clarity
4. Line 162: what do the reported ranges in brackets represent?
5. Line 329 - 331: this last sentence, from my reading, seems to contradict the above data. Check and revise for clarity.
6. Line 380: you need to make it very clear here that all these statements are made relative to the control conditions. That was not apparent to me until later in this section.
7. Line 384: I am ignorant of assays for *C. elegans*, but is this fluorescent assay monitoring intracellular glutathione concentrations?
8. Line 487 - 489: confusing sentence, please revise for clarity
9. Figure 5: in your legend define what the "+ media" conditions are. This is unclear
10. Figure 6: I think this may be an oversight: was the Ag+Te vs Te comparison significant?
11. Table S1. Are the concentrations shown the MIC of the silver and tellurite during individual exposure? Or are they the amounts that were added in combination to achieve synergistic inhibition? Also, isn't "FIC/FBC {greater than or equal to}0.8 and {less than or equal to}1.2=indifferent" just "additive"?

Staff Comments:

Preparing Revision Guidelines

Please return the manuscript within 60 days; if you cannot complete the modification within this time period, please contact me. If you do not wish to modify the manuscript and prefer to submit it to another journal, please notify me of your decision immediately so that the manuscript may be formally withdrawn from consideration by Microbiology Spectrum.

Thank you again for submitting your paper to Microbiology Spectrum.

Response to reviewers comments Spectrum

Dear Editor and reviewers.

Below follows direct responses to each of the reviewers comments. In addition to the changes made in response to the reviewer comments, some further edits were made particularly in the final 'clean version' where many typographical and grammar issues were corrected by having 4 others read this final version. We also noticed that we had descriptions of some methods written in both the main Text and the Supplementary. The Supplementary has now been edited to only include methods of experiments that are either indicated in the main Text, but described in Supplementary, or for experiments only reported in Supplementary. Finally reference numbers were corrected in the final unmarked 'clean' version.

You will see we have taken the reviewers comments seriously editing the manuscript to in line with their comments.

On Behalf of the authors we, we thank you for this opportunity to revise our manuscript and have it reconsidered

Prof. R. J. Tuner

Reviewer comments:

Reviewer #1 (Public repository details (Required)):

RNA-seq data needs to be uploaded to GEO

Reviewer #1 (Comments for the Author):

In this work Pormohammad, A et. al used transcriptional, metabolomics and biochemical assays to understand the toxicity of silver and tellurite when administered together as an strategy to kill *Pseudomonas aeruginosa*. The joint administration of these metals is more toxic for bacteria than the exposure to metals alone in laboratory strains and clinical isolates. Global transcriptional changes are reported which seem to be correlated with metabolomic and biochemical data. Even when this work is interesting due to the need to improve the current antibacterial therapies, there are many aspects that need to be improved as will be mentioned.

Minor comments

The method to determine MBC from cultures previously used to assess MIC might not been very accurate since a 1:15 dilution is not high enough to get rid of the antibiotic in cultures exposed to higher concentrations of drugs (which may inhibit bacterial growth). Since antibacterials were not washed away and authors do not mention the range and dilutions (related to MIC) of concentrations used it is not evident to determine if the values of MBC are correct. A more accurate method to determine MBC is to determine the number of CFU counts after treatment with different concentrations of antibacterials.

We employed various methods to determine the MBCs, namely: 1. Using a multipin replica platter to transfer 2 ul to or pipette to transfer 10 ul from the MIC determination 96 well plate to a plate of fresh media which dilutes the remaining antimicrobial. This gives a dilution of the original concentration down below MIC levels for all but the highest concentrations; 2. Similar to method 1 but transferring to an agar plate to observe recoverable colonies; 3. Transferring 20 ul to a plate for CFU counts. We used all these methods and we did not see any difference in results. Methods 1 and 2 are less costly and thus we used them more routinely while method 3 was used occasionally for validations. We have expanded on the text in the methods for further clarification.

The way to start cultures needs to be specified. It is not evident if the 1.0 McFarland culture was generated from an overnight culture or from colonies scrapped from a plate.

We have clarified the approach in the Supplementary methods in MIC section that the inoculant was prepared from a plate. We have a defined CFU inoculant that is now stated.

In figure 1 it would make much easier for reader to analyze data if the average of biological replicates (repeats) were shown into only one colored square/number.

The goal was to show via the heat map the variation of the susceptibility. The values are defined in the heat map legend. We purposely did not show the average and standard deviation in order to honestly show the variability between biological trials of the experiment. By reporting values in a Table it would be impossible to appreciate this variability...something that has led to misunderstandings and false dogma in the literature. According to our reasoning, no change has been made.

RNA-seq data needs to be uploaded to GEO and an accession number has to be included in this manuscript.

*We missed to submit to the archive. It is now submitted. To review GEO accession GSE232832: Go to <https://www.ncbi.nlm.nih.gov/geo/query/acc.cgi?acc=GSE232832> Enter token **yjafmiashfijun** into the box*

All the experiments in this manuscript were conducted by exposure of cultures to metals alone or in combination for 2 hours, except for those showing the effect of antimicrobial of membrane integrity (Figure 7). Authors need to explain why they did that, if they want to compare with their other results, they need to be consistent.

Sorry, this was our mistake. This experiment was done like all the others that is growing to OD600 of 0.08 (1.5×10^8 CFU/ml) and then challenged with the metals and/or antibiotics for a further 2 hours. Our mistake is to have transferred the text of the methods of a thesis without checking the experimental conditions used here. Thus, all data shown, is for 2 hrs of challenge as this time frame little difference in overall cell density between challenged and unchallenged cultures were seen.

It is not evident for this reviewer the point that authors want to state with the experiments using *C. elegans*. A better description may improve the understanding of these results and why they need to be included in this manuscript.

A few sentences and related citation are now added to the start of the results section (2.5) of this work.

Major concerns

The transcriptional profile experiment compares exponentially growing cultures treated with Ag, Te or Te+Ag, with a PBS treated culture that keep growing and will eventually reach stationary phase (implying a completely different metabolic and transcriptional status). Cultures treated with metals alone or in combination (at MIC) will stop growing. It is not accurate to say that bacteria "is responding" to a treatment if the initial condition is not known (as mentioned throughout the whole manuscript), so a most proper control would be to compare the transcriptional status post treatment with metals to the one that bacteria display before treatment (prior to the addition of toxicants). Otherwise, language needs to be softened and modified as "global transcriptional changes" instead of cell/bacterial response. Also, since in stationary phase there is a lack of nutrients which generates a global stress situation (oxidative stress included), these results might underestimate their findings when cultures are challenged with metals. Authors should discuss this possibility in the text.

See below our explanation that relates here as these two issues are interconnected. The cells did not reach stationary phase, the OD600 of 0.08 is just below mid log phase, a further 2 hrs did not see entrance to stationary phase, which from our growth curves would have been at ~3.5 hrs. The bacteria became stressed during the 2 hrs of incubation but did not stop growing. Our conditions were in part chosen in that

there would be little cell density differences at end of challenge time. For the experiments of physiological stress we always had a PBS treated control for comparison.

However, I do agree that we should modify our wording as we did not do single cell 'omics and thus must recognize we have a community that by definition would be a range of physiological states. Owing to this, we have modified this wording throughout.

Since authors are comparing the total amount of metabolites present in cultures exposed to different antimicrobial compounds, experiments shown in Figures 5, S6, S7 need to be normalized ideally by CFU counts or by total protein content, especially since differences between experimental conditions are subtle (2-6-fold). Even at the MIC those compounds will be killing cells differently when alone or in combination, so a 2-6 difference in the total amount of cells in the assay may imply that all the conditions display no differences with each other. For this reason, it is not possible to compare cultures challenged with metal(oids) alone or in combination without proper normalization.

Unfortunately, we failed here to make it clear and obvious in how we wrote our methods. The transcriptomic and metabolomics samples were taken from the same culture. We evaluated the CFU and after 2 hrs all cultures were determined to be $1.5 (+/- 1.5) \times 10^8$ CFU/ml for all samples. Thus, indicating that the chosen Ag and Te concentrations did not reduce growth during the 2 hr exposure (although metal exposed cultures were found to be non-viable after 8 hrs).

This is a very important comment and is worthwhile to extend further explanation here. It is important to remember that MIC's are determined from an experiment where a low cell density is added to media and one defines an end point after extended incubation 16-24hrs, so even if there was some growth at the beginning before the stress overwhelmed and thus it would not be visible in such assays. Additionally, the antimicrobial load is much higher per cell at the start of cultures (moles of antimicrobial per cell) compared to what occurred in our experiments. The goal was not to kill the cells but to stress them to understand their response. If we had killed the cells, there would be no RNA as mRNA has very short half-lives (time frames of minutes at most for most genes). Thus, the stress must be high enough yet still allow transcriptional and metabolic changes to occur.

The methods section has been changed to improve understanding of sample collection and the degree of normalization of cultures.

A good example of this is the comparison of H₂O₂ levels shown in Figure 6 when authors compare levels to negative control (untreated cultures) which after 2 hours growing will reach stationary phase containing more than 100X more cells than the treated conditions. Those differences in the total amount of cells used in the assay are not displayed in the experiments of this manuscript.

Like the transcriptomic and metabolomic experiments, cell culture density was similar at point of assay. The unchallenged culture did not overgrow the challenged as described above. This has been clarified in methods.

In figure 8 authors suggest that combined administration of silver-tellurite stimulates the health of the animal. However, in the experimental design they mention "The *C. elegans* wild-type strain N2 (Bristol) and *E. coli* OP50 at 10⁹ CFU/mL as a food source were used in this study (25, 26). *E. coli* and 10 times the MIC of each antibacterial component were added to the agar plates." Since *E. coli* OP50 will be facing lethal concentrations of metals is it possible that *C. elegans* prefers death or dying cells to grow? Authors should discuss this possibility.

*This is a very good observation. We did not observe crowding but we agree that during this experiment the *C. elegans* did run out of food. A quick test while working on these revisions shows that at the 10 times MIC concentrations the *E. coli* have lost viability by day 2 but are not lysed. So, the worms could continue to feed on the dead cells, but since they are not viable the worms would run out of food by end of the experiments. As noted in observations in day 4 they start new generations, so this lack of 'live' food did not affect them at least in the time frame of this experiment. Since we used the two antibiotics groups (Ciprofloxacin and Gentamicin in 10 times MIC concentration) which led to the same effect we could compare the results between antibiotics and the metals. It remains that the worms seem to be more active*

and healthier under the combined Ag+Te metal stress which is our take home message. We have added statements to the methods recognizing this issue. On the other hand, we do not want to comment if the worms are supposed to prefer live over dead bacteria for growth (or so says the published articles. But perhaps they choose the bacteria that are pickled with metals. Regardless, a few sentences to recognize this was added to the methods in the main text.

Brighter rod-shaped structures are observed in Ag-Te treated worms in Figure S8. Are these bacteria? If so, this observation may imply that Ag-Te administration kills E. coli OP50 generating oxidative stress as has been reported by other groups. These findings may support the previous comment suggesting that it is easier for C. elegans to be fed by dead bacteria.

The brighter structures the reviewer is indicating are not bacteria as scale of these are much larger than ~0.5 cm in size compared to bacteria (0.0015 cm). The oval shaped structures are embryo's; the longer rodlike structures may strange and may be hatched larvae. Comment on this has been added to the results Supplementary material. We do not want to put this in the main text as we don't want the effects to the animal model distracting from the focus of the effects on the bacteria (at least for this manuscript).

Reviewer #2 (Comments for the Author):

This is a very interesting work that adds to both basic scientific knowledge on the subject, and also has potential applications. I have a few recommendations for the paper:

1) I would recommend moving the sound from the supplementary videos and fixing the titles - while watching the videos, I could hear a recording of a conversation in the background. Some of the titles are not fully visible in the video, and there is a typographical error in at least one of them. The captions could also indicate what the arrows are pointing out.

We agree that the quality of these videos is not of "high-quality" production. Unfortunately, we had limited resources and videos were made by a camera phone against the microscope lens. However, we are confident that they still provide a reasonably useful visual for the text. We have edited the captions.

2) There are a number of grammatical errors/words missing or incorrect, etc. in the manuscript, for example at lines 114-115; 293-294; 298-302; 312; 345; 355; 394; 400; 435-439; 444-445; 458;465-467; 499; 522-523; 638; 640-642; 658; and in the captions for Figures 4, 6, and 9. There are also a number of errors in the supplementary material including a few incorrect words. There are errors in the titles for Figures S8 and S9 as well. I would recommend having someone go over the manuscript for these types of errors, as that will make it easier to read.

These and other minor errors have been identified and corrected throughout the manuscript. Several sentences throughout have also had their grammar and wording reworked for clarification of their statements, particularly in discussion.

3) There is a problem with reference 98 in the main manuscript, as written, and also with references 5 and 27 in the supplementary material - I would recommend carefully reviewing the references and ensuring they have all been entered correctly.

Through this revision we have identified a few issues, the pain with reference managers and not checking. All references have been checked and several new ones added, and others deleted. We have also streamlined the references in the Supplementary to those only unique to this section.

4) Figures S1 and S2 have very small print. It would be better to spread them out over a few pages so that they are easier to read.

As these are supplementary and a file that can be downloaded, interested parties can enlarge to capture specific data if interested. An important aspect is the heat map characteristics that if spread out would be lost from view. No change made.

5) If the tellurite is just enhancing the mechanisms of action of silver treatment, in situations where silver resistance develops (i.e. to those mechanisms of action), presumably adding tellurite would not help? Perhaps the authors could address this.

This comment is probably the result of some of our statements (lines 467 to line 473 in original) which are not clear enough. As we wrote in the revised version, " We have two findings: 1. silver and tellurite target different systems, yet some of these systems overlap considerably. 2. Co-application of silver-tellurite targets several systems in the same way as the individual metal exposure but leads to a more sever effect on those systems. This means the silver with tellurite combination doesn't target different mechanisms for their synergistic antimicrobial activity, they simply magnify each other's effects." (lines 480-485).

Overall, these changes are quite minor, in an otherwise extensive and informative paper.

Thank you for your comments

Reviewer #3 (Public repository details (Required)):

Transcriptome raw reads should be deposited in SRA or equivalent database. Submitting expression data to NCBI GEO or the EMBL equivalent would also be nice as it would enhance the extractability/usability of the dataset.

We missed to submit to the archive. It is now submitted.

To review GEO accession GSE232832:

Go to <https://www.ncbi.nlm.nih.gov/geo/query/acc.cgi?acc=GSE232832>

*Enter token **yjafmiashfijtun** into the box*

Reviewer #3 (Comments for the Author):

This manuscript describes a systems-level characterization of the synergism between tellurite and silver in their toxicity towards *Pseudomonas aeruginosa*. Studies of metal(loid) mixtures, in general, is an emerging area in the microbiology world. I believe that both clinical and environmental microbiology researchers would find interest in this paper and I am enthusiastic to see it published after revisions. The authors initially show nice synergism between the two metal(loids) in various *P. aeruginosa* clinical isolates. They then go on to characterize the transcriptome, metabolome, and other biochemical responses of a laboratory strain of *P. aeruginosa* to the combined treatment relative to the individual exposures. I really liked the integration of the physiological data at the end with the transcriptomic data. These data fit together nicely. In contrast, the metabolomics data was not well integrated at all, and I was left wondering why it was even included since it felt very "isolated" from the other data. Further work needs to be done to better integrate these data into the story. Finally, the paper needs thorough English language editing from either a fluent co-author, a trusted fluent colleague, or a professional editing service. Specific comments follow:

Major:

1. It looks like I am missing the supplemental excel tables with all the gene expression and metabolome data. Please include those in the next submission. Also, consider depositing your raw reads in the SRA (or equivalent). Gene expression data would also be more accessible if deposited in an archive like NCBI GEO.

We missed to submit to the archive. It is now submitted.

To review GEO accession GSE232832:

Go to <https://www.ncbi.nlm.nih.gov/geo/query/acc.cgi?acc=GSE232832>

*Enter token **yjafmiashfijtun** into the box*

2. This manuscript represents one of a relatively limited number studies that have begun looking at metal(loid) mixtures on a systemic level (i.e., using omics-based approaches) in microorganisms. Since this is such an emerging and exciting area of research, it may be useful to look for connections between your work and those other studies. This would be much more interesting and novel than the extended descriptions of individual tellurite and silver toxicity in the Discussion sections-topics that have been very well-reviewed many times over. For examples, are there certain emergent properties of all metal mixtures regardless of composition? Or is each mixture its own unique "thing"? In overlap in differentially expressed pathways? Even tying in some of the senior author's own prior work on metal(loid) mixture impacts on microbial physiology within the discussion would be extremely interesting/informative. Some suggested papers:

Dey, Priyadarshini, et al. "Insight into the molecular mechanisms underpinning the mycoremediation of multiple metals by proteomic technique." *Frontiers in Microbiology* 13 (2022).

Obviously this first paper is on a fungus (*Aspergillus*), but I think there may actually be some overlap with the data you highlight here

Goff, Jennifer L., et al. "Mixed heavy metal stress induces global iron starvation response." *The ISME Journal* 17.3 (2023): 382-392.

Bacillus and under anaerobic conditions (so may have some significant differences due to that...hard to judge, though, since I don't have the complete transcriptome dataset)

Brun, Nadja R., et al. "Mixtures of aluminum and indium induce more than additive phenotypic and toxicogenomic responses in *Daphnia magna*." *Environmental science & technology* 53.3 (2019): 1639-1649.

This last one would be more relevant to the metal mix effects in your nematode system. And, in general, there are a lot more studies out in the multicellular eukaryotic world on systems-level metal mixture effects that could be drawn upon to add context to your nematode observations.

We are very thankful for bringing these papers to our attention. Our literature search concentrated on the metal-based antimicrobials and not on 'heavy metals'. We had seen one but not the others. Being aware of comment # 8 (see below) telling us that the Discussion was a bit too 'review-y', we were hesitant to expand to discuss other studies but feel it might be worthwhile to have a sentence on each to state the common thread to ours. So, we have added a paragraph in this regard as the second last in the Discussion. If anything, it provides a case to continue to study mixed metal exposure to variety of organisms.

3. Line 128: For the uninitiated, it might be useful to state if *C. elegans* is a useful model for pre-human testing toxicology studies. Otherwise, I am left wondering, why are you studying metal toxicity in a nematode.

*We have added a sentence regarding this in the introduction but also a preamble to section in results and a citation of a review of use of *C. elegans* as a toxicology tool.*

4. Line 132: At the beginning of your results, you need to explicitly state that all work (*work?*) was conducted under aerobic conditions as oxygenation can significantly impact the toxicity of metal(loid)s

Yes, a statement of aerobic conditions is now added in these early sentences of results.

5. Line 166: You need to lead this section by immediately discussing exposure conditions. So, move section 2.2.1 up top. You can include the PCA details after you introduce the exposure conditions.

Moved as suggested.

6. Section 2.3: Can you better integrate the metabolomic data with the transcriptome data. You do a beautiful job of this in Section 2.4. I would love to see something similar done with this section. For example, do any DEGs correlate with metabolite changes? Is there a figure you could create of metabolic pathways to highlight the integration of the data? Although, I realize this is extracellular data so maybe that's not feasible.

This reviewer is correct, that it is not easy/possible to link information from different omics experiments. Particularly because the response times are so different with metabolites changing on time frame of 10's of

seconds and transcript profiles changing on time scale s 10-100 that. Regardless we have made thoughts towards relationships to the data. In the results have now more specifically noted a possible relationship between Xanthine and the nucleotides and signaling molecules that relate to systems around general stress regulation (ppGpp) and biofilm (c-di-GMP) (lines 286-287). Additionally, In the amended version, (lines 255-275) some of the metabolomic data are now integrated with the transcriptomic results. We examined, in particular, the presence of succinate in the growth medium of cells treated with the duo silver/tellurite. We note this now in our additional summary cartoon Fig 10. These data are also commented in the Discussion section from line 492-495.

7. Line 346: tellurite reduction to Te by glutathione will also release superoxide. So, there may be multiple ROS sources.... With that said, do you see evidence of tellurite reduction in your cultures?

Not all bacterial species generate ROS with tellurite exposure even though most all see a reduction in RSH content due to the oxidation of glutathione in the reduction. We thank the reviewer to remind us as such observations were not initially recorded. Upon repeating the experiment for this revision, we observed an visible agreement to the RSH data of Fig S6 where the tellurite exposed culture turned black but the Ag-Te culture was only light grey (from far less Tellurium nanomaterial). We have now added this observation to section 2.4.2 sulfur homeostasis.

8. Discussion paragraphs 2 and 3: as already mentioned above, these paragraphs were too long and "review-y". They really took me out of the story since you aren't tying it to any of your results. I imagine they could be easily condensed to a couple of sentences each briefly stating (1) major mechanisms of individual metal(loid) toxicity and (2) other mixtures synergism has been observed in.

We agree that these paragraphs are quite review-y, but we feel that is key background information for readers less familiar on either silver or tellurium antimicrobial activities. We imagine readers from the silver field not knowing tellurium and vice versa. We had this initially in the Introduction, but it did not read well there either. Regardless we have reduced the length of these two paragraphs by a few sentences each by removing some redundant comments and some rearrangement to improve readability. Since we have added now an additional summary cartoon figure (new Fig. 10), this information is good comparator background.

9. Line 504: if Ag exposure alone is increasing actP expression, then how would there be an increase in tellurite uptake? There is simply no tellurite in the system to be taken up. Perhaps this is just a confusing sentence that needs re-wording

*Your comment is absolutely correct. The entire sentence has been deleted and rephrased as follows:
"We observe that the actP1 gene of P. aeruginosa is upregulated by silver but strongly downregulated by the Ag/Te duo. Thus, the co-presence of tellurite generates a sort of rescue effect which represses its entry via ActP1"*

10. In the final couple paragraphs of the discussion you present a model to describe the synergism between the two metals. I can see where its going, but it is very hard for me to visualize in my head. Could you prepare a figure to illustrate this?

Yes, this is our first study of its kind and we did not expect such an avalanche of data. Now we believe to have a good overview but we do agree it is still hard to grasp. We have added an additional figure (Fig 10) to summarize all together the best we can at this research stage.

11. Figure 1: very visually busy figure that would probably be better as a table with averaged values (+/- SD). You could potentially still color the table concentration

The actual values are less important than the trend of susceptibility. The goal was to show via the heat map the variation of the susceptibility. The values are defined in the heat map legend. We also purposely did not show the average and standard deviation in order to honestly show the variability between biological trials of the experiment. By reporting values in a Table it would be impossible to appreciate this variability...something that has led to misunderstandings and false dogma in the literature. No change has been made.

12. Most figures: where relevant, I would like to see exposure concentrations given on figures or in the legends (but preferably on the figures themselves)

Exposure concentrations of the metal/metalloids as well as the antibiotics are now added in the captions of figures in both the main Text and Supplementary.

13. Figure 8: again, this is an instance where a table summarizing these data may be more helpful. You can simply average your replicate groups. Showing each individual replicate make the table too big and challenging to read. However, other possibilities include transforming into individual scatter plots and only showing the most interesting ones in the main text

Here again similar to figure 1 we wanted to be honest with showing the variation between biological trials and a visual that displays the differences quickly to the reader. This heat map illustrates well what we wanted the reader to catch from the data. The individual numbers are less important and thus we generated a scoring of "advantageous" to the organism or not. A change has been made in the text to point to this figure in this regard.

14. Just a comment relating to future studies, but have you considered working with the organotellurium(IV) drug AS101? It has been used as an antimicrobial (among other things) in mammalian systems so its pharmacological profile is pretty well-characterized and may be more feasible for trials in mammalian systems (if that is your goal).

Thank you very much for your suggestion. The AS101 was explored for tumor treatment is where obviously there is greater interest and greater investment. There are actually now six organo-tellurium compounds reported to have some form of useful biological response. There is also the possibility of methylated tellurium forms having activity...considered in earlier texts to be the reason for antioxidant properties of elephant garlic which has a high content of Me_n-Te.

Minor:

1. There is inconsistent capitalization of chemical names (particularly tellurite, silver, and metabolites mentioned later in the text). These should be lowercase unless, of course, they are the first word of the sentence.

I consider the names of the elements proper noun and should be capitalized, but I see the point that most do not, so corrected to lower case throughout.

2. Line 80: "The Lancet" should be italicized

Corrected, thanks

3. Line 160: Are these MICs/MBCs really the same for all the clinical isolates? Table 1 seems to suggest not. Please edit for clarity

This has been clarified to indicate the proportion of clinical isolates the concentrations were effective at.

4. Line 162: what do the reported ranges in brackets represent?

Defined as range of values found for clinical isolates.

5. Line 329 - 331: this last sentence, from my reading, seems to contradict the above data. Check and revise for clarity.

Sentences from the previous section on H₂O₂ have been removed and the section on Fe release been reworked to improve clarity. The reading is more consistent and less convoluted to show both H₂O₂ and Fe are consistent for Se-Te exposure.

6. Line 380: you need to make it very clear here that all these statements are made relative to the control conditions. That was not apparent to me until later in this section.

Corrected - defined earlier.

7. Line 384: I am ignorant of assays for *C. elegans*, but is this fluorescent assay monitoring intracellular glutathione concentrations?

The term 'cellular' has been added to clarify. There would be some extracellular GSH but is typically already oxidized so would not influence the results.

8. Line 487 - 489: confusing sentence, please revise for clarity

Paragraph has now been reworked, see amended version, from line 499-503: "Therefore, the net loss of membrane potential caused by the production of superoxide anions through electron leakage, mostly at the level of the uncoupled ETC, is possibly compensated by the electrochemical gradient linked to SOO activity (Fig. 10). Based on this proposal, the mechanism triggered by tellurite is seen as a turbo system since it recycles a waste/unwanted product (O_2^-) limiting the overall losses of the process."

9. Figure 5: in your legend define what the "+ media" conditions are. This is unclear.

These controls are now more explicitly explained in the methods section. But also now defined in legend.

10. Figure 6: I think this may be an oversight: was the Ag+Te vs Te comparison significant?

Thanks for pointing this out, we had the lines between these two bars, but removed as it made the figure sloppy. We have now added the statistical information to the figure caption as well as to the sentence in section 2.4.2.

11. Table S1. Are the concentrations shown the MIC of the silver and tellurite during individual exposure?

Or are they the amounts that were added in combination to achieve synergistic inhibition? Also, isn't "FIC/FBC {greater than or equal to}0.8 and {less than or equal to}1.2=indifferent" just "additive"?

A sentence added to footnote to define concentration are those of each metal that gave the synergistic load. As for FIC values, we have seen 1.2 cut off to be called both indifferent or additive. Changed to include both terms.

May 31, 2023

Prof. Raymond J Turner
University of Calgary Department of Biological Sciences
Biological Sciences
Biological Sciences Building
2500 University Dr NW
Calgary, Alberta T2N 1N4
Canada

Re: Spectrum00628-23R1 (Insights into the Synergistic Antibacterial Activity of Silver Nitrate with Potassium Tellurite Against *Pseudomonas aeruginosa*)

Dear Prof. Raymond J Turner:

Thank you for submitting your manuscript to Microbiology Spectrum. As you will see your paper is very close to acceptance. Please modify the manuscript along the lines I have recommended. As these revisions are quite minor, I expect that you should be able to turn in the revised paper in less than 30 days, if not sooner. If your manuscript was reviewed, you will find the reviewers' comments below.

When submitting the revised version of your paper, please provide (1) point-by-point responses to the issues raised by the reviewers as file type "Response to Reviewers," not in your cover letter, and (2) a PDF file that indicates the changes from the original submission (by highlighting or underlining the changes) as file type "Marked Up Manuscript - For Review Only". Please use this link to submit your revised manuscript. Detailed instructions on submitting your revised paper are below.

Link Not Available

Sincerely,

Paolo Visca

Reviewer comments:

Reviewer #3 (Comments for the Author):

The author's have significantly revised the manuscript and have addressed all of my prior concerns. I very much appreciate their careful consideration of the comments. I am excited to see this paper published in the near future! A couple of very very very minor comments follow to improve the clarity of the final manuscript:

1. Lines 141, 149 (and elsewhere): change to "reference strains". "Indicator strains" generally refers to indicators of fecal contamination in water.
2. Line 154-155: Define FIC and FIB here since the methods section is at the end.
3. Line 265-271: state the condition(s) these genes are differentially expressed under (I am assuming just Te-Ag...)
4. Line 439: "Te" rather than "Te0" (the latter is the elemental form and you are just discussing tellurium generally here).
5. Line 495: typo "sever"
6. Line 560: awkward sentence, please revise
7. One of the reviewers had comments on the growth phases at sample collection times. While the author's methodology was apparent to me, I could see how readers may have similar confusion. I would suggest including an example growth curve in the

supplemental information and marking out the sampling time (in addition to the new explanation in the results section).

Preparing Revision Guidelines

Please return the manuscript within 60 days; if you cannot complete the modification within this time period, please contact me. If you do not wish to modify the manuscript and prefer to submit it to another journal, please notify me of your decision immediately so that the manuscript may be formally withdrawn from consideration by Microbiology Spectrum.

Response to reviewers on version Spectrum00628-23R1

Corrections are marked with yellow highlighter in 'revised_marked' version.

Reviewer #3 (Comments for the Author):

The author's have significantly revised the manuscript and have addressed all of my prior concerns. I very much appreciate their careful consideration of the comments. I am excited to see this paper published in the near future! A couple of very very very minor comments follow to improve the clarity of the final manuscript:

1. Lines 141, 149 (and elsewhere): change to "reference strains". "Indicator strains" generally refers to indicators of fecal contamination in water.
I was going by the nomenclature used in the ATCC documentation, but I agree. We have changed the wording throughout.
2. Line 154-155: Define FIC and FIB here since the methods section is at the end.
Sentence changed to "FIC (fractional inhibitory concentration) scores ranged from 0.51 to 0.16 and FIB (fractional eradication concentration) from 0.62 to 0.157 defining synergy for the silver-tellurite combination against all clinical isolates (Synergy calculations in Supplementary material and data in Table S1)."
3. Line 265-271: state the condition(s) these genes are differentially expressed under (I am assuming just Te-Ag...)
It is referring to both Te and Ag-Te, ie the effect of Te. Sentence changed to "This latter observation is interesting and supports two further results under both the tellurite challenges, namely:....."
4. Line 439: "Te" rather than "Te0" (the latter is the elemental form and you are just discussing tellurium generally here).
Yes thanks changed to (Te)
5. Line 495: typo "sever"
Corrected to 'severe'
6. Line 560: awkward sentence, please revise
Sentence revised to "Within the metal toxicity field, we see studies of metal exposure to many different lifeforms and the data tends complex with surprises being unmasked in mixed metal exposure studies."
7. One of the reviewers had comments on the growth phases at sample collection times. While the author's methodology was apparent to me, I could see how readers may have similar confusion. I would suggest including an example growth curve in the supplemental information and marking out the sampling time (in addition to the new explanation in the results section).
An example growth curve has now been added to the supplementary. As well the following sentence has been added to materials and methods under Sample Preparation. "The example growth curve in Fig S9 reflects that over the time frame of the metal challenge, the bacteria became stressed but did not stop growing and cell harvest

occurred before loss of cell significant cell density would be a concern in the comparison of data to the unchallenged controls. “

June 5, 2023

Prof. Raymond J Turner
University of Calgary Department of Biological Sciences
Biological Sciences
Biological Sciences Building
2500 University Dr NW
Calgary, Alberta T2N 1N4
Canada

Re: Spectrum00628-23R2 (Insights into the Synergistic Antibacterial Activity of Silver Nitrate with Potassium Tellurite Against *Pseudomonas aeruginosa*)

Dear Prof. Turner:

Your revised manuscript has now been accepted, and I am forwarding it to the ASM Journals Department for publication. You will be notified when your proofs are ready to be viewed.

Sincerely,

Paolo Visca
Editor, Microbiology Spectrum
